

# Ammonium CI-Orbitrap: a tool for characterizing the reactivity of oxygenated organic molecules

Dandan Li[1], Dongyu Wang[2], Lucia Caudillo[3], Wiebke Scholz[4], Mingyi Wang[5,6], Sophie Tomaz[1], Guillaume Marie[3], Mihnea Surdu[2], Elias Eccli[4], Xianda Gong[7], Loic Gonzalez-Carracedo[8], Manuel Granzin[3], Joschka Pfeifer[3,9], Birte Rörup[10], Benjamin Schulze[6], Pekka Rantala[10], Sébastien Perrier[1], Armin Hansel[4], Joachim Curtius[3], Jasper Kirkby[3,9], Neil M. Donahue[5], Christian George[1], Imad El-Haddad[2], Matthieu Riva[1,*]

[1] Univ Lyon, Université Claude Bernard Lyon 1, CNRS, IRCELYON, 69626, Villeurbanne, France

[2] Laboratory of Atmospheric Chemistry, Paul Scherrer Institute, 5232, Villigen, Switzerland

[3] Institute for Atmospheric and Environmental Sciences, Goethe University Frankfurt, 60438, Frankfurt am Main, Germany

[4] Institute for Ion Physics and Applied Physics, University of Innsbruck, 6020, Innsbruck, Austria

[5] Center for Atmospheric Particle Studies, Carnegie Mellon University, Pittsburgh, PA, 15213, USA

[6] Division of Chemistry and Chemical Engineering, California Institute of Technology, Pasadena, CA 91125, USA

[7] Leibniz Institute for Tropospheric Research, 04318, Leipzig, Germany

[8] Faculty of Physics, University of Vienna, Vienna, 1090, Austria

[9] CERN, the European Organization for Nuclear Research, CH-1211 Geneve 23, Switzerland

[10] Institute for Atmospheric and Earth System Research/Physics, Faculty of Science, University of Helsinki, 00014, Helsinki, Finland

[*] Email: matthieu.riva@ircelyon.univ-lyon1.fr





Abstract
Oxygenated organic molecules (OOMs) play an important role in the formation of atmospheric
aerosols. Due to various analytical challenges in measuring organic vapors, uncertainties remain in the
formation and fate of OOMs. The chemical ionization Orbitrap mass spectrometer (CI-Orbitrap) has
recently been shown to be a powerful technique able to accurately identify gaseous organic
compounds due to its great mass resolving power. Here we present the ammonium ion ($NH_4^+$) based
CI-Orbitrap as a technique capable of measuring a wide range of gaseous OOMs. The performance of
the CI-($NH_4^+$)-Orbitrap was compared with that of state-of-the-art mass spectrometers, including a
nitrate ion ($NO_3^-$) based CI coupled to an atmospheric pressure interfaced to long time-of-flight mass
spectrometer (APi-LTOF), a new generation of proton transfer reaction-TOF mass spectrometer
(PTR3-TOF), and an iodide ($I^-$) based CI-TOF mass spectrometer equipped with a Filter Inlet for
Gases and AEROsols (FIGAERO-CIMS). The instruments were deployed simultaneously in the
Cosmic Leaving OUtdoors Droplets (CLOUD) chamber at the European Organization for Nuclear
Research (CERN) during the CLOUD14 campaign in 2019. Products generated from α-pinene
ozonolysis across multiple experimental conditions were simultaneously measured by the mass
spectrometers. $NH_4^+$-Orbitrap was able to identify the widest range of OOMs (i.e., O ≥ 2), from low
oxidized species to highly oxygenated volatile organic compounds (HOM). Excellent agreements were
found between the $NH_4^+$-Orbitrap and the $NO_3^-$-LTOF for characterizing HOMs and with the PTR3-
TOF for the less oxidized monomeric species. A semi-quantitative information was retrieved for
OOMs measured by $NH_4^+$-Orbitrap using calibration factors derived from this side-by-side
comparison. As other mass spectrometry techniques used during this campaign, the detection
sensitivity of $NH_4^+$-Orbitrap to OOMs is greatly affected by relative humidity, which may be related to
changes in ionization efficiency and/or multiphase chemistry. Overall, this study shows that $NH_4^+$ ion-
based chemistry associated with the high mass resolving power of the Orbitrap mass analyzer can
measure almost all-inclusive compounds. As a result, it is now possible to cover the entire range of
compounds, which can lead to a better understanding of the oxidation processes.





## 1 Introduction

Aerosols affect the climate by either directly scattering or absorbing solar radiation, or acting as seeds for cloud formation (Fan et al., 2016; Haywood and Boucher, 2000). A major fraction of submicron aerosol mass consists of organic compounds, with secondary organic aerosol (SOA) predominating (Jimenez et al., 2009; Hallquist et al., 2009). Oxygenated organic molecules (OOMs) generated from the oxidation of volatile organic compounds (VOCs) contribute to the formation and growth of SOA (Ehn et al., 2014; Mellouki et al., 2015). OOMs can be generated by the autoxidation of peroxy radical ($RO_2$) (Bianchi et al., 2019), where $RO_2$ undergoes an intramolecular H atom shift, followed by $O_2$ addition forming a new and more oxidized $RO_2$ (Crounse et al., 2013; Rissanen et al., 2014; Bianchi et al., 2019). These propagation steps are repeated until termination occurs by either bimolecular or unimolecular reactions, yielding closed-shell molecules (Bianchi et al., 2019). Among the OOMs, the highly oxygenated organic molecules (HOMs), containing multiple functional groups and exhibiting (extremely) low saturation vapor pressure, can nucleate in concert with inorganic species e.g., sulfuric acid or on their own (Ehn et al., 2014; Kirkby et al., 2016; Bianchi et al., 2016), forming new particles. Less oxygenated molecules (i.e., containing 2 to 5 oxygen atoms) play a vital role in the growth of newly formed atmospheric particles, either by condensation or through multiphase chemistry (Bianchi et al., 2019; Ehn et al., 2014; Hallquist et al., 2009). Therefore, the identification and quantification of the wide diversity of OOMs are essential to understand SOA formation and growth (Kirkby et al., 2016; Bianchi et al., 2016; Trostl et al., 2016; Jokinen et al., 2015; Glasius and Goldstein, 2016).

Mass spectrometry (MS) has made remarkable achievements in detecting, characterizing, and quantifying OOMs (Wang et al., 2017; Wang et al., 2020; Breitenlechner et al., 2017; Bianchi et al., 2019; Ehn et al., 2010; Riva et al., 2019a; Breitenlechner et al., 2017). Moreover, the application of chemical ionization (CI) enables the detection of a wide variety of organic and inorganic analytes (Bianchi et al., 2019; Ehn et al., 2014; Jokinen et al., 2012; Lee et al., 2014). However, the selection of ionization chemistry in combination with MS detection technique will impact the methods selectivity and sensitivity toward certain groups of OOMs (Bianchi et al., 2019; Riva et al., 2020; Riva et al., 2019b; Berndt et al., 2018b; Berndt et al., 2018a). For example, negative ion-based chemistry, including nitrate ($NO_3^-$), can optimally detect HOMs, which only constitute a small subset of the OOMs (Lee et al., 2014; Berndt et al., 2015; Berndt et al., 2018b; Riva et al., 2019b). Positive ion-based chemistries have also been developed, showing great sensitivity to HOMs as well as less oxidized products, providing the possibility of achieving carbon closure of the OOMs (Praplan et al., 2015; Berndt et al., 2018a; Berndt et al., 2018b; Hansel et al., 2018; Riva et al., 2020; Riva et al., 2019b). However, these positive ion methods are mainly based on proton transfer and often result in fragmentation of the analytes (Yuan et al., 2017; Breitenlechner et al., 2017; Li et al., 2020). Time-of-flight (TOF) mass spectrometers using ammonium ($NH_4^+$) or amines as reagent ions can detect a wide variety of OOMs but suffer from a lack of mass resolving power, making peak identification challenging, especially for complex systems, i.e., under ambient conditions (Berndt et al., 2018b; Berndt et al., 2018a; Riva et al., 2019b). Finally, the recently developed Orbitrap mass spectrometer using propylamine has achieved unambiguous identification of overlapping peaks and accurate





quantification of OOMs (Riva et al., 2019a). However, this analytical technique has been used in very
diluted and dry environments to ensure a linear response to the OOMs produced from simple
atmospheric systems, i.e., a single VOC precursor and oxidant (Riva et al., 2020; Riva et al., 2019b).

Here, we explore the capability of $NH_4^+$ ion-based CI-Orbitrap mass spectrometer (Q-Exactive

Orbitrap, Thermo Scientific) for detecting OOMs generated from α-pinene ozonolysis in the Cosmic
Leaving OUtdoors Droplets (CLOUD) chamber at the European Organization for Nuclear Research
(CERN) under various environmental conditions. We compare the performance of the CI-($NH_4^+$)-
Orbitrap to state-of-the-art online mass spectrometers including a nitrate CI atmospheric pressure
interface long time of flight mass spectrometer (CI-($NO_3^-$)-APi-LTOF; Tofwerk AG), a proton transfer
reaction time of flight mass spectrometer (PTR3-TOF; Ionicon Analytik GmbH), and an iodide CI
time of flight mass spectrometer equipped with a Filter Inlet for Gases and AEROsols (I$^-$-FIGAERO-
CIMS, Tofwerk AG).

## 2 Experimental approach and product analysis

### 2.1 CLOUD chamber experiments

All experiments were conducted in the CLOUD chamber, a 26 m$^3$ cylindrical stainless-steel vessel at
CERN. The chamber can achieve a pristine background for the study of nucleation (Kirkby et al.,
2011; Kirkby et al., 2016). The chamber operated as a continuously stirred tank reactor (CSTR), with
mixing driven by two inductively coupled fans at the top and bottom of the chamber. Evaporated
liquid nitrogen ($N_2$) and liquid oxygen ($O_2$) were blended at a ratio of 79:21 to provide ultra-pure
synthetic air, which flushed the chamber constantly. Variable amounts of trace gases, including $O_3$,
VOCs, $NO_x$, $SO_2$, and CO were accurately injected into the chamber via a gas control system and
monitored. Photolysis was driven by various light sources, including Hg-Xe UV lamps, and UV
excimer laser. Between experiments, the chamber was cleaned by irrigating the walls with ultra-pure
water, then heated to 373 K, and flushed with humidified pure air and high ozone, reducing the
contaminant (e.g., VOCs) to sub pptv levels. During the cleaning process, particles were removed
using a high-voltage electric field.

The results presented here were from the CLOUD14 campaign performed in autumn 2019.

During CLOUD14, the total flow was kept at 250 standard liters per minute (slpm), providing an
average residence time of 104 minutes. α-Pinene was introduced into the chamber by passing a small
flow of dry air over a temperature-controlled evaporator containing liquid α-pinene. Ozone was
generated by flowing a small fraction of the air through a quartz tube surrounded by UVC lights
(wavelength < 240 nm). Experiments were performed at low temperature (263 ± 0.1 K). The RH in the
chamber was controlled by flowing a portion of the air through a Nafion® humidifier using ultrapure
water (18 MΩ cm, Millipore Corporation). The contents of the chamber were monitored by a wide
range of external instruments connected to the sampling probes that protrude ~1 m into the chamber.





## 2.2 Product analysis by CI-(NH$_4^+$)-Orbitrap

The chemical composition of closed-shell molecules was determined in real time by means of a CI-Orbitrap sampling from the CLOUD chamber through a 750 mm long, 10 mm inner diameter Teflon tube at a flow rate of 10 slpm. The CI inlet mounted on the Orbitrap was custom-built with minor modifications from the commercial inlet (Riva et al., 2019a). The ion-molecule reaction (IMR) proceeded at atmospheric pressure with a residence time of 200-300 ms. The same operating parameters used in our previous studies (RF level 60, automatic gain control $1 \times 10^6$ charges, maximum injection time 1000 ms, multi RF ratio 1.2, mass resolution m/$\Delta$m 140,000 at $m/z$ 200), were used, thereby minimizing declustering and maximizing the linearity range (Riva et al., 2019a; Riva et al., 2020; Cai et al., 2022).

The high resolution Orbitrap mass spectra data were analyzed using "Orbitool" software with a graphical user interface (GUI) (https://orbitrap.catalyse.cnrs.fr) (Cai et al., 2021). The analysis procedures included data averaging, noise determination and reduction, single peak fitting, mass calibration, assignment of molecular formulas, and export of time series. Signals were averaged over 5 min before determining the noise and performing mass calibration.

NH$_4^+$ has been utilized in proton-transfer reaction mass spectrometry (PTR-MS) (Lindinger et al., 1998; Berndt et al., 2018b; Hansel et al., 2018). Here, this ionization technique was used to detect OOMs and was operated in a similar fashion as in our initial study (Riva et al., 2019a). NH$_3$ was added into the ion source by flushing 2 sccm of dry air over the headspace of a 1% liquid ammonia water mixture (prepared from a MilliQ water and a 25% ammonium hydroxide stock solution, ACS reagent, Sigma-Aldrich). The product molecules ("prod") were softly charged by binding to ammonium (NH$_4^+$) ions, forming (prod)-NH$_4^+$ adduct ions or protonated products (prod)-H$^+$, following either reaction (1) or (2),

$$\text{NH}_4^+ + \text{prod} \rightarrow \text{(prod)-NH}_4^+ \tag{1}$$

$$\text{NH}_4^+ + \text{prod} \rightarrow \text{(prod)-H}^+ + \text{NH}_3 \tag{2}$$

Direct detection of the NH$_4^+$ reagent ion cannot be detected due to the cut-off of the Orbitrap mass analyzer (i.e, m/Q 50). Due to the absence of a reagent ion signal normalization of the raw analyte signal is difficult and hinders quantification of OOMs. However, we observed a total of 62 peaks corresponding to amines, including C$_4$H$_{12}$N$^+$, and C$_6$H$_{14}$N$^+$, which are formally ammonia derivatives. To some extent, their signals can be used to correct for changes in NH$_4^+$ ion chemistry. Among these peaks, 13 were abundant and constant throughout the measurement period (Fig. S1). As a result, these signals were used as surrogates for the primary reagent ion signal to normalize the signal intensity of the analytes and to account for the potential variation of the ionization process (Riva et al., 2019b).

The concentrations of OOMs, including monomeric species such as C$_{10}$H$_{14}$O$_x$ and C$_{10}$H$_{16}$O$_x$, were estimated based on correlation analysis between NH$_4^+$-Orbitrap and NO$_3^-$-LTOF/PTR3-TOF for species with the same elemental composition, providing semi-quantitative measurements. Compounds





with a Pearson correlation coefficient greater than 0.9 were used to estimate the concentration of the
compounds measured by the $NH_4^+$-Orbitrap according to the following equation:
$$[prod]_{Orbi\_LTOF/PTR3} = c_{Orbi\_LTOF/PTR3} \times \frac{[(prod)-NH_4^+] + [(prod)-H^+]}{\sum[Amine]}$$  (3)
where $c$ is the calibration factor, obtained from the correlation analysis. Finally, a temperature-
dependent sampling-line loss correction factor was applied (Simon et al., 2020).

**2.3 Product analysis by CI-$(NO_3^-)$-APi-LTOF**


Detection of $RO_2$ radicals and closed-shell products was also performed by the CI-$(NO_3^-)$-LTOF
which has been described elsewhere (Jokinen et al., 2012; Kurten et al., 2014). Therefore, only
relevant details for this study are provided here. The $NO_3^-$-LTOF used in this study had a mass
resolving power of m/Δm 12,000 and detected OOMs (mass 300-650 Da) as clusters ions with
$(HNO_3)_n(NO_3^-)$ anions, with n = 0-2. The primary ions were produced by a corona discharge needle
exposed to a sheath gas enriched by $HNO_3$. Laminar flow diffusional loss was assumed in the 30 cm
sampling line. A core-sampling technique was applied (Knopf et al., 2015), which drew a core flow of
5.1 slpm from the center of a 30 slpm total flow. This setup reduced the sampling loss rate of HOMs to
less than 30% (Simon et al., 2020). The data were processed using Tofware (Version 3.2, Aerodyne
Inc., USA) and MATLAB R2019b (MathWorks, Inc., USA). In addition, background signals, mass-
dependent transmission efficiency (Heinritzi et al., 2016), and sampling losses (Simon et al., 2020)
were determined and corrections were applied. The $NO_3^-$-LTOF was directly calibrated using sulfuric
acid (Kurten et al., 2012). A calibration factor C was determined to be $\sim4.13 \times 10^{10}$ molecules cm$^{-3}$
during CLOUD14 (Caudillo et al., 2021).

**2.4 Product analysis by PTR3-TOF**


The PTR3-TOF ionizes organic compounds by proton transfer where $H_3O^+$ ions were produced by a
corona discharge using humidified nitrogen (Breitenlechner et al., 2017). To reduce sample losses, a 2
slpm was drawn from a 10 slpm laminar flow through a critical orifice into the tripole where the ion-
molecule reactions occur. The pressure in this region was maintained at ~80 mbar. The distribution of
primary ions and sample molecules can be adjusted by a tunable radio frequency signal applied to the
tripole rods.

During the CLOUD14 experiments, the collision energy was controlled between 62 and 72 Td to

reduce the methods humidity dependence which may complicate the detection of organic compounds.
The PTR3-TOF was calibrated using a gas standard mixture containing 1 ppm of 3-hexanone,
heptanone, and α-pinene in nitrogen. The concentration of oxygenated products was estimated using
the sensitivity of 3-hexanone as lower-limit values due to possible fragmentation. All data were
analyzed using TOF-Tracer software running on Julia 0.6 (https://github.com/lukasfischer83/TOF-
Tracer) and were further corrected for the duty cycle transmission of TOF and temperature dependent
sampling line losses (Stolzenburg et al., 2018).





**2.5 Product analysis by I⁻-FIGAERO-CIMS**
The I⁻-FIGAERO-CIMS was capable of characterizing both gas and particle phases (Lopez-Hilfiker et
al., 2014). In the gas-phase mode, gases were directly sampled into a 100-mbar turbulent ion-molecule
reactor, while particles were collected onto a polytetrafluoroethylene (PTFE) filter through a separate
dedicated sampling port. Analytes were then ionized with I⁻ chemical ionization and extracted into a
TOF mass analyzer (Wang et al., 2020). In this study, only gas phase data are reported.
Iodide ions (I⁻) were used as the reagent ions and formed by passing a 1.0 slpm flow of ultrahigh
purity $N_2$ over a diffusion tube filled with methyl iodide ($CH_3I$), and then through a $^{210}Po$ radioactive
source. In the sampling mode, the reagent ion flow was mixed with a sample flow in the IMR at ~150
mbar. Coaxial core sampling was used to minimize the vapor wall loss in the sampling line. The total
flow was kept at 18.0 slpm and the core flow at 4.5 slpm; the instrument sampled at the center of the
core flow with a flow rate of 1.6 slpm. The gas-phase background signal was determined by routinely
introducing zero air directly into the inlet. Data were analyzed using Tofware (2.5.11_FIGAERO
version; Aerodyne Inc., USA) giving 10 s average mass spectra. The ion signal was normalized by the
sum of reagent ion signals (i.e., m/Q 127: I⁻ and 145: $H_2OI^-$).
**2.6 Volatility of OOMs**
It is challenging to directly measure the vapor pressure of individual OOMs due to the difficulty to
acquire authentic standards. To overcome experimental challenges, model calculations have been
developed to estimate the vapor pressure using, for example, structure-based estimations and formula-
based estimations (Pankow and Asher, 2008). Volatility basis set (VBS), a categorization framework
based on quantifiable organic property (i.e., volatility) has been established and is frequently used to
characterize oxidation chemistry (Donahue et al., 2011; Li et al., 2016). The VBS parameterization is
useful for classifying the wide range of OOMs into multiple volatility groups, including extremely low
volatility organic compounds (ELVOC) and low volatility organic compounds (LVOC) based on their
effective saturation concentration ($C^*$) in the unit of µg m⁻³ (Bianchi et al., 2019). In this study, we
applied the VBS parameterization optimized by Li et al (Li et al., 2016; Isaacman-Vanwertz and
Aumont, 2021).
$$log_{10}C^*(298K) = \left(n_C^0 - n_C\right)b_C - n_O b_O - 2\frac{n_C n_O}{(n_C+n_O)}b_{CO} - n_N b_N - n_S b_S \qquad (4)$$

where $n_C$, $n_O$, $n_N$, and $n_S$ was the number of carbon, oxygen, nitrogen, and sulfur atoms of the
specific molecule, separately; $n_C^0$ was the reference carbon number; $b_C$, $b_O$, $b_N$, and $b_S$ was the
contribution of each atom to $log_{10}C^*$, respectively; $b_{CO}$ was the carbon-oxygen nonideality (Donahue
et al., 2011). Values of $b$ coefficient can be found in Li et al.(Li et al., 2016). The formula used to
estimate the vapor pressure was amended to convert all $NO_3$ groups into OH groups to reduce the bias
from the compounds containing nitrates (Daumit et al., 2013; Isaacman-Vanwertz and Aumont, 2021).





Due to the different temperatures in the CLOUD14 experiments, we adjusted $C^*(298K)$ to the
measured experimental temperature in equations (5) and (6):

$$log_{10}C^*(T) = log_{10}C^*(298K) + \frac{\Delta H_{vap}}{Rln(10)} \times (\frac{1}{298} - \frac{1}{T}) \qquad (5)$$

$$\Delta H_{vap}(kJ\ mol^{-1}) = -11 \cdot log_{10}C^*(298K) + 129 \qquad (6)$$

where $T$ was the temperature in Kelvin, $C^*(298K)$ was the saturation vapor concentration at 298 K,
$\Delta H_{vap}$ was the evaporation enthalpy and R was the gas constant (8.3134 J K$^{-1}$ mol$^{-1}$). The potential
presence of isomers may result in uncertainty in this method since the only input is the compound's
molecular formula.
In this study, all oxidation products were grouped into six volatility regimes; ultralow-volatility
(ULVOCs, C$^*$ < 10$^{-8.5}$ μg m$^{-3}$), extremely low volatility (ELVOCs, 10$^{-8.5}$ < C$^*$ < 10$^{-4.5}$ μg m$^{-3}$), low-
volatility (LVOCs, 10$^{-4.5}$ < C$^*$ < 10$^{-0.5}$ μg m$^{-3}$), semi-volatile (SVOCs, 10$^{-0.5}$ < C$^*$ < 10$^{2.5}$ μg m$^{-3}$),
intermediate-volatility organic compounds (IVOC, 10$^{2.5}$ < C$^*$ < 10$^{6.5}$ μg m$^{-3}$), and VOC (10$^{6.5}$ < C$^*$ μg
m$^{-3}$) based on VBS.
**3 Results and Discussions**
**3.1 Characterization of NH$_4^+$-Orbitrap**
First, the ability of the higher mass resolving power of the Orbitrap for separating overlapping mass
spectral peaks was compared to other TOF mass analyzers (Riva et al., 2019a; Riva et al., 2020). The
identification and quantification of peaks with low intensity were most affected by overlapping
signals, therefore, the relative intensities of neighboring peaks should also be considered when
estimating their ease of separation.
The mass resolving power was defined as

$$\text{mass resolving power} = m\ /\ \Delta m \qquad (7)$$

where $m$ was the mass-to-charge ratio of the analyte ion, and $\Delta m$ was the full width at half maximum
(FWHM). Higher mass resolving power allows unambiguous mass spectral peak assignment. For a
pair of overlapping peaks of equal intensity, the distance between their respective peak center, referred
to hereafter simply as peak distance, $dm$, needed to be greater than approximately 0.8 of the FWHM of
the overlapping peaks, such that they could be reasonably deconvolved as shown in Fig. S2.
Depending on their experience, individuals may be able to visually identify the presence of
overlapping peaks at lower or higher $dm$ values. We arbitrarily defined the minimum $dm$ (normalized
to that of FWHM, or $\Delta m$) as the value at which the observed spectrum ("Combined" trace in Fig. 1
and S2) had a local minimum between the centers of the overlapping peaks (i.e., there was a "dip" in
the observed signal between ion peaks). The minimum $dm$ value increased with the intensity ratio of
overlapping peaks, ranging roughly from 0.85 (for equally intense peaks) to 1.43 (for peaks differing
one order of magnitude in their respective intensities), as shown in Fig. 1. In practice, noise and the





presence of additional neighboring peaks would further complicate peak deconvolution. For
simplicity, we used a normalized $dm$ of 1 (i.e., $dm = \Delta m$) as a threshold for unambiguous
deconvolution of neighboring peaks.
Figure 2 shows the histogram of the distances between neighboring peaks normalized against the
FWHM for the Orbitrap mass analyzer and a TOF analyzer having a mass resolving power of 10,000.
In each histogram, one count indicated that an ion had at least one neighboring ion with a relative
intensity of 20%, 50%, or 100% (with a higher relative intensity threshold value being less selective).
Neighboring ions separated by distances exceeding 2 times the FWHM were considered well-
separated. For ions with multiple neighboring peaks within the 2 x FWHM separation distance
window, the distance to the first neighboring peak that satisfied the aforementioned relative intensity
threshold was reported. Overall, $NH_4^+$-Orbitrap can separate most of the observed ions (> 99%), while
the CI-($NO_3^-$)-TOF, depending on the relative intensity threshold set, can separate only 32% to 46% of
all the ions by at least 1 FWHM.
**3.2 Characterization of OVOC by four instruments**
Illustrated in Fig. 3 are Kendrick mass defect plots of OOMs measured by CI-($NH_4^+$)-Orbitrap, CI-
($NO_3^-$)-LTOF, PTR3-TOF, and I$^-$-FIGAERO-CIMS, identifying species of 484, 252, 145, and 67,
respectively. The $NH_4^+$-Orbitrap detected the widest range of products, including HOMs and the less
oxidized species (i.e., O < 6). Out of the 484 compounds, 5% were amines. The number of O atoms in
OOMs varied from 1 to 11 in monomers ($C_2$-$C_{10}$) and from 2 to 16 for dimeric products ($C_{14}$-$C_{20}$), with
an average elemental oxygen-to-carbon ratio (O:C) of 0.4 ± 0.2. As expected, the $NO_3^-$-LTOF
exhibited a very good sensitivity towards HOMs, with the highest O:C of 0.7 ± 0.3. The PTR3-TOF
mainly detected compounds below m/Q 300 Th with an average O:C of 0.5 ± 0.3. However, the
PTR3-TOF was optimized (i.e., lowering E/N value) to achieve sensitive measurements of ammonia
and amines, impacting its capability to detect OOMs. Gas-phase concentrations of OOMs were too
low to be efficiently detected in real-time via the gas-phase sampling port of the I$^-$-FIGAERO-CIMS.
As a result, fewer monomers of $C_{8-10}$ and dimers of $C_{19-20}$ were observed, with an average O:C of 0.5 ±

0.2.

**3.3 Instrumental comparisons: correlations**
Due to differences in selectivity and sensitivity of the analytical methods toward OOMs, only a
fraction (~22%) of the identified species can be compared among two or more mass spectrometers. To
identify how $NH_4^+$-Orbitrap performed compared to the other mass spectrometers, a correlation
analysis including all co-detected ions was complied. The experimental conditions of the runs used for
performing this analysis are summarized in Table S1. The data set covered a variety of conditions,
such as different concentrations of α-pinene, $NO_x$, $SO_2$, and CO, as well as RH. A Pearson correlation
coefficient $R^2$ was calculated, using the time series of OOMs having the same elemental composition
measured by the different mass spectrometers. Figure 4 displays the correlation coefficient of detected
compounds, with marker size scaled by $R^2$. The $NH_4^+$-Orbitrap showed $R^2$ > 0.80 for ~30% of the



compounds co-detected with the $NO_3^-$-LTOF. In addition, it demonstrated high correlations with
PTR3-TOF including 89 species having a $R^2 > 0.50$. $I^-$-FIGAERO-CIMS showed $R^2 > 0.90$ for certain
families of compounds, including $C_{10}H_{15}O_{5-7}N$ and $C_{20}H_{31}O_{7,9}N$. By comparing the coverage regions
of the instruments across multiple experimental conditions, the $NH_4^+$-Orbitrap was capable of covering
the widest range of compounds and showed an overall good agreement with other mass spectrometers.
**3.4 Instrumental comparisons: concentration estimates**
Concentrations of the identified compounds were estimated for $NH_4^+$-Orbitrap, as described in section
2.2. The sensitivity of $NH_4^+$-Orbitrap was constrained using semi-quantitative information from the
other instruments. For instance, concentrations of the most abundant $C_{10}$-monomers (i.e., $C_{10}H_{14/16}O_n$)
were estimated using different calibration factors (Fig. 5), which were measured during steady-state
conditions (i.e., Run 2211 with $[O_3] = 100$ ppbv and $[\alpha$-pinene$] = 2$ ppbv, RH = 10%). Using
calibration factors obtained from two independent correlation analyses (i.e., using the PTR3-TOF and
the $NO_3^-$-LTOF), concentrations of $C_{10}$-monomers measured by $NH_4^+$-Orbitrap were within a factor of
2, showing a good agreement between the different instruments and underlining the robustness of the
methodology. As previously reported, the Orbitrap had a non-linear response to compounds present at
extremely low concentrations, which was independent of the sample composition, instrumental setup,
or the reagent ion (Riva et al., 2020; Cai et al., 2022). A similar evaluation was performed for the
$NH_4^+$-Orbitrap by comparing the measured versus the theoretical isotopic intensities. As shown in Fig.
S3, the $NH_4^+$-Orbitrap had a linear response for ion intensity greater than $\sim5 \times 10^3$ cps, which
corresponded to a limit of quantification (LoQ, corresponding to the lowest normalized signal
observed within the linear range) of $\sim5 \times 10^5$ molecules $cm^{-3}$ for OOMs, estimated using the
calibration factor derived from $NO_3^-$-LTOF; which is consistent with a previous study (Riva et al.,

2020).

Figure 6 presents the concentrations of all OVOCs measured by the $NH_4^+$-Orbitrap determined by

applying two different calibration factors. Compared to other mass spectrometers, the $NH_4^+$-Orbitrap
captured the largest fraction of the reacted carbon. Of the most abundant oxidation products,
pinonaldehyde (i.e., $C_{10}H_{16}O_2$) was not efficiently detected by $NO_3^-$-LTOF, which is consistent with
the higher selectivity of the $NO_3^-$ reagent ion. To further illustrate the selectivity of the different
reagent ions, Fig. 7 offers a summary of the performance of each mass spectrometer in detecting
monomeric compounds, such as $C_{10}H_{16}O_n$. The y-axis is arbitrary and represents a qualitative
characterization of the oxygen content when compounds were detected by different CI schemes.
Similar to previous results, the $I^-$-FIGAERO-CIMS detected OOMs with $n_O > 3$ , but was not optimal
for the detection of monomers with $n_O > 7$ (Riva et al., 2019b). The $NO_3^-$-LTOF was mainly selective
towards HOMs with $n_O > 6$ (Riva et al., 2019b). The PTR3-TOF had limited capabilities in detecting
OOMs with $n_O > 5$ due to the optimization of the instrument to obtain a very sensitive measurement of
ammonia. Previously, the amine-CI demonstrated promise for the detection of OOMs, but was limited
to applications with comparatively clean conditions due to considerable depletion of the reagent ion
and the presence of overlapping peaks (Berndt et al., 2017; Berndt et al., 2018b; Riva et al., 2019b).





While showing a similar OOMs detection range to amine-CI, $NH_4^+$-CI in tandem with the greater mass
resolving power of the Orbitrap mass analyzer provided a linear response to higher loading. As shown
in Fig. S4, background peaks were not affected by atmospherically relevant concentrations of $O_3$ and
α-pinene. Overall, the $NH_4^+$-Orbitrap appears to have the potential for providing a more reliable
identification/quantification of OOMs produced from VOC oxidation compared to other existing mass
spectrometry techniques.

**3.5 Volatility distribution by four instruments**

Figure 8 shows the distribution of oxidation products measured by four MS instruments according to
their saturation vapor concentrations ($\log_{10}C_{sat}$) estimated using the modified Li et al. approach (Li et
al., 2016; Isaacman-Vanwertz and Aumont, 2021). OOMs were grouped into six volatility regimes
based on a volatility basis set (VBS): ultra-low volatility (ULVOCs); extremely low volatility
(ELVOCs); low-volatility (LVOCs); semi-volatile (SVOCs); intermediate volatility organic
compounds (IVOC); and VOC. ULVOCs and ELVOCs initiate cluster growth and form new particles.
The total signal in each volatility bin represented the sum of the signal intensity of OOMs within the
volatility range. The mean contributions of these compound regimes are shown in the VBS pie charts.
The ULVOC, ELVOC, and LVOC regimes were well captured by $NH_4^+$-Orbitrap and $NO_3^-$-LTOF.
The PTR3-TOF only characterized the SVOC and IVOC regime (along with VOCs). IVOC and VOC
regimes in the PTR3-TOF and $NH_4^+$-Orbitrap were generally less oxygenated VOCs (i.e., $n_O < 5$).
IVOC comprised the biggest mass contributions for the $NH_4^+$-Orbitrap, and LVOC dominated in the
$NO_3^-$-LTOF. Hence, the detection of the $NH_4^+$-Orbitrap covered the widest range of volatilities,
clearly highlighting the benefit of using this technique for the formation and fate of OOMs. In the past,
reagent switching has not been practical, and users would run multiple mass spectrometer systems in
parallel or use a Multi-scheme chemical IONization inlet (MION) with only one mass spectrometer to
obtain the fullest possible mass spectrum (Rissanen et al., 2019; Huang et al., 2021). With $NH_4^+$-
Orbitrap it is now possible to cover the entire range of compounds which was not the case with most
CI techniques.

**3.6 RH dependance of $NH_4^+$-Orbitrap**

The sensitivity of the reagent-adduct ionization has been reported to be affected by the presence of
water vapor for a variety of reagent ions (Lee et al., 2014; Breitenlechner et al., 2017). The impact of
RH on the detection of OOMs by the $NH_4^+$-Orbitrap was also studied. While the concentrations of gas
phase precursor and oxidant remained constant the RH was raised from 10% to 80%. During this
increase the signal of organic vapor behaved inconsistently, presumably due to the increased
partitioning of SVOCs to the particle phase under an otherwise constant gas-phase production rate
(Surdu et al., 2023) and an increase in the condensation sink (Fig. S5). As shown in Fig. 9, the $NH_4^+$-
Orbitrap demonstrated an RH dependence. For instance, the signal of less oxygenated molecules (i.e.,
$n_O < 5$) increased with increasing RH, especially compounds with $n_C = 8$; while the signal of highly
oxygenated molecules (i.e., $n_O > 10$) decreased as a function of RH. The average behavior of all $C_{8-10}$
monomers and $C_{18-20}$ dimers was summarized and compared between four instruments (Fig. S6). The


other three mass spectrometers also showed obvious RH dependence. The evolution of the signal
intensity of the ions measured by the $NH_4^+$-Orbitrap with changing RH may be explained by multiple
reasons, such as water affecting the ionization efficiency or altering the physicochemical processes of
the gas phase chemistry.
First, the efficiency of a particular compound partly relied on whether water vapor competes with
the ammonium ion, lowering the sensitivity, or whether it acted as a third body to stabilize the
ammonium-organic analyte cluster by removing extra energy from the collision, raising the sensitivity
(Lee et al., 2014). $NH_4^+$ primary ions can cluster with water molecules when humidity increased,
thereby reducing the clustering of the $NH_4^+$ with organic analytes (Breitenlechner et al., 2017).
However, the formed $NH_4^+X_n$ (X being $NH_3$ or $H_2O$; n = 1,2) clusters might also act as reagent ions
and ionized OOMs through ligand switching reactions, which were expected to be fast and thus
improve the charging efficiency (Hansel et al., 2018). Compared to previous $NH_4^+$-CIMS, the $NH_4^+X_n$
reagent ions were expected to be larger due to the absence of the field in the ion-molecular-reaction
zone in Orbitrap, resulting in greater ligand exchanging and increasing the sensitivity for the less
oxygenated species (Canaval et al., 2019). For RH-independent compounds, this may be due to the
existence of very stable complexes with $NH_4^+$ reagent ion, or sufficient internal vibrational modes to
disperse extra energy from the collision (Lee et al., 2014). The highly oxygenated dimers in the
category of ULVOCs and ELVOCs which largely partition to the particle phase regardless of the
presence of water might indicate that water may also affect the physicochemical processes (i.e.,
multiphase chemistry, partitioning, etc.), in this case possibly causing the decomposition of highly
oxygenated molecules in the particle phase to create less and moderately oxygenated products, e.g.,
$C_8H_{12}O_{1-5}$ (up to a 30-fold increase in the gas phase) (Pospisilova et al., 2020), and/or leading to an
increase in the driving force of gas-particle partitioning of highly oxygenated species (Surdu et al.,
2023). Finally, while water vapor could affect the gas-phase chemistry through water reactions with
the Criegee intermediates (CIs), $HO_2$ chemistry, OH radical concentration, no clear evidence has been
identified as earlier discussed by Surdu et al (2023).
**4 Summary**
In conclusion, this study presented an intercomparison between CI-($NH_4^+$)-Orbitrap, CI-($NO_3^-$)-LTOF,
PTR3-TOF, and $I^-$-FIGAERO-CIMS based on the identification and quantification of OOMs formed
from the ozonolysis of α-pinene under various environmental conditions. We used for the first time,
$NH_4^+$ adduct ions with the Orbitrap mass spectrometer to measure the oxygenated species. $NH_4^+$-
Orbitrap was a promising CIMS technique for a comprehensive measurement of the whole product
distribution and provides a more complete understanding of the molecular composition and volatility
of OOMs. This allows $NH_4^+$-Orbitrap to better monitor the evolution of organic compounds, which
can be beneficial for air quality, pollutant transport, and climate models. However, it remains
challenging to obtain an accurate quantification of the trace gaseous substances at pptv levels. It is
worth expecting that $NH_4^+$-Orbitrap can be not only useful for laboratory-based studies but also to
field observations, to provide a deeper understanding of atmospheric oxidation processes.



**Conflicts of interest**

At least one of the (co-)authors is a member of the editorial board of Atmospheric Measurement Techniques

**Acknowledgements**

We thank the European Organization for Nuclear Research (CERN) for supporting CLOUD with important technical and financial resources. We thank the Orbitool team for developing the tools to analyze mass spectra. This work was financially supported by the French National program LEFE (Les Enveloppes Fluides et l'Environnement), the European Research Council (ERC-StG MAARvEL; no. 852161), the European Union's Horizon 2020 research and innovation programme (Marie Sklodowska-Curie grant agreement no. 764991 and 701647), the Swiss National Science Foundation (no. 200021_169090, 200020_172602, 20FI20_172622, and 206021_198140), the US National Science Foundation (NSF_AGS_1801280, NSF_AGC_1801574, NSF_AGS_1801897, NSF_AGS_2132089), and the German Federal Ministry of Education and Research (CLOUD-16 01LK1601A). D.D.L. thanks the China Scholarship Council of P. R. China for the Ph.D. grant. M.Y.W. acknowledges financial support from the Schmidt Science Fellows Program by Schmidt Futures, in partnership with the Rhodes Trust.

**Author Contributions**

D.D.L., D.Y.W., L.C., W.S., M.Y.W., S.T., G.M., M.S., E.E., X.D.G., L.G.-C., M.G., J.P., B.R., B.S., P.R., S.P., A.H., J.C., J.K., N.M.D., C.G., I.E.-H., and M.R. prepared the CLOUD facility or measuring instruments. D.D.L., D.Y.W., L.C., W.S., M.Y.W., S.T., G.M., M.S., E.E., X.D.G., L.G.-C., M.G., J.P., B.R., B.S., J.K., and M.R. collected the CLOUD data. D.D.L., D.Y.W., L.C., W.S., M.Y.W., G.M., and M.R.and analysed the data. D.D.L., D.Y.W., M.Y.W., N.M.D., C.G., I.E.-H., and M.R. wrote the manuscript and contributed to the scientific discussion. All authors discussed the results and commented on the paper.

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

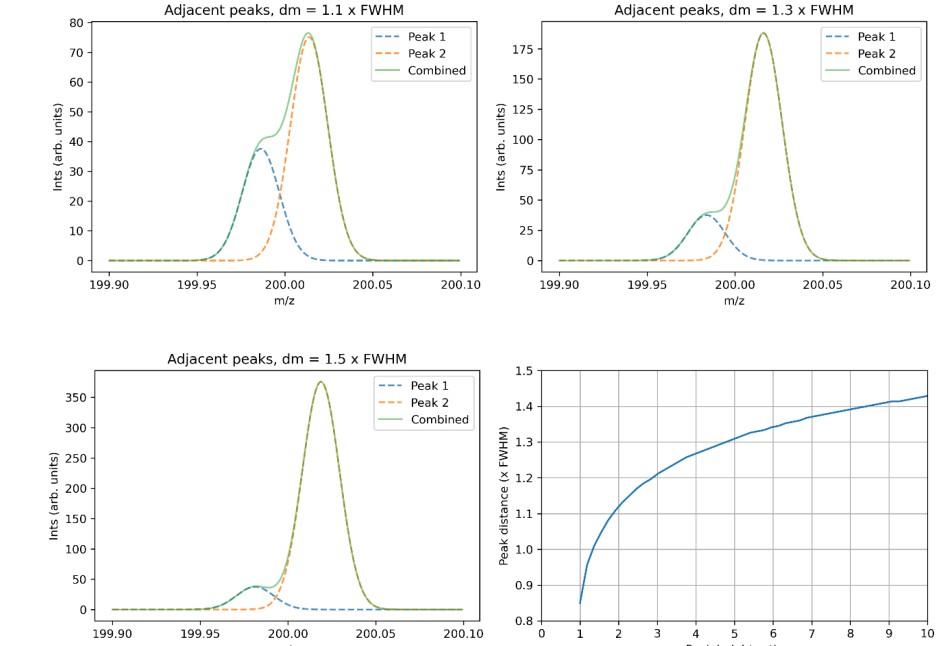

**Figure 1:** Simulated TOF spectra of overlapping peaks of equal intensity near *m/z* 200 assuming a mass resolving power of 8,000, somewhere between that of a Tofwerk HTOF ("high-resolution time-of-flight") and LTOF ("long high-resolution time-of-flight) mass spectrometers. The overlapping area represents a greater proportion of the peak area of the less intense peak. Noise wasn't added to the data.



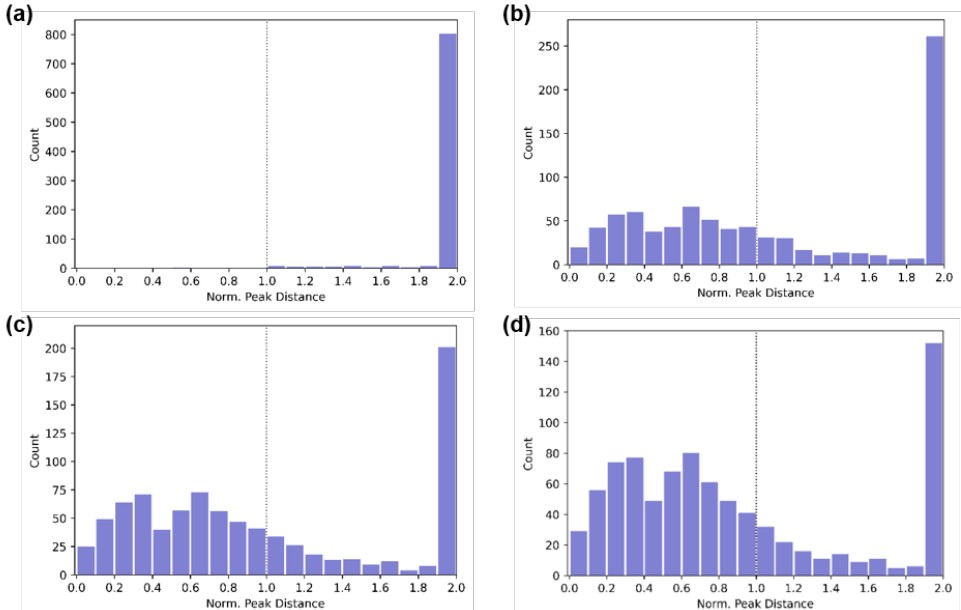

**Figure 2:** Distance of neighboring peaks normalized to the FWHM, which is a function of the mass resolving power of the mass analyzer and the nominal mass of the ion. For each ion, the distance to the closest neighbor with a relative peak intensity that exceeds 20%, 50%, or 100% is recorded. The average mass spectra observed during Run 2213 by the $NH_4^+$-Orbitrap is used to generate the analysis. (a) Orbitrap mass analyzer (mass resolution ∼140,000) >99% of ions are separated by at least 1 FWHM from their neighbors with relative intensity threshold being set at 20%. (b) TOF mass analyzer (mass resolution ∼10,000) >46% of ions are separated by at least 1 FWHM from their neighbors with a relative intensity threshold being set at 100%. (c) TOF mass analyzer >39% of ions are separated by at least 1 FWHM from their neighbors with a relative intensity threshold is set at 50%. (d) TOF mass analyzer >32% of ions are separated by at least 1 FWHM from their neighbors with a relative intensity threshold is set at 20%.

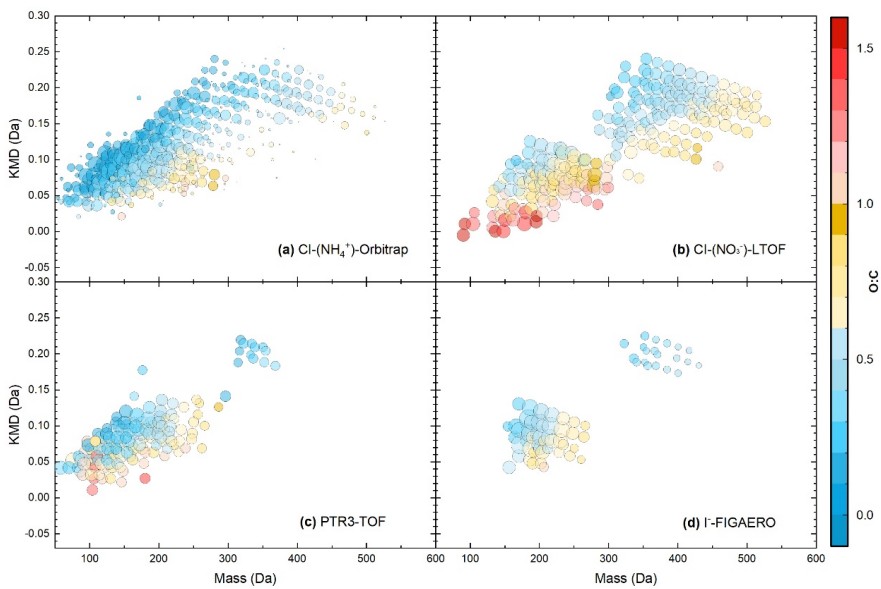

740

**Figure 3:** Mass defect plots for organic compounds measured by (a) CI-(NH$_4^+$)-Orbitrap, (b) CI-(NO$_3^-$)-LTOF, (c) PTR3-TOF and (d) I$^-$-FIGAERO-CIMS in run 2211. The x-axis represents the mass-to-charge ratio of the neutral analyte and the y-axis represents the corresponding mass defect, which is the difference between their exact mass and nominal mass (Schobesberger et al., 2013). Markers were all sized by the logarithm of their corresponding signals and colored by the O:C value.





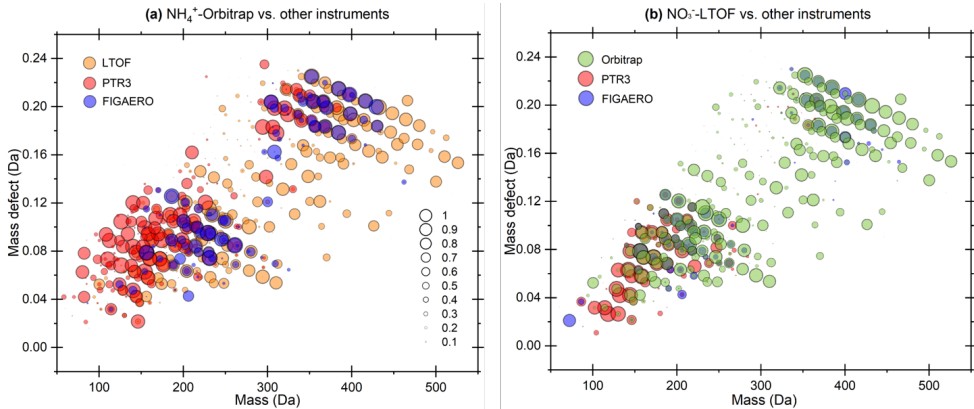

**Figure 4:** Mass defect plots depicting the compounds of which time series correlation was observed by (a) CI-($NH_4^+$)-Orbitrap and (b) CI-($NO_3^-$)-LTOF with other MS instruments. Each circle represents a molecule and marker size represents the correlation $R^2$ of time series of the molecule between two different MS instruments. Two sets of data in run 2211 and 2213 were used to reduce uncertainties.





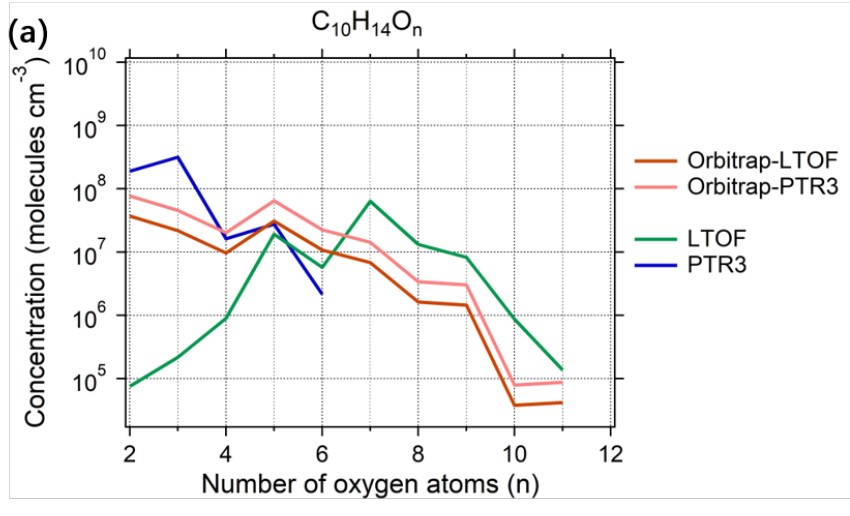

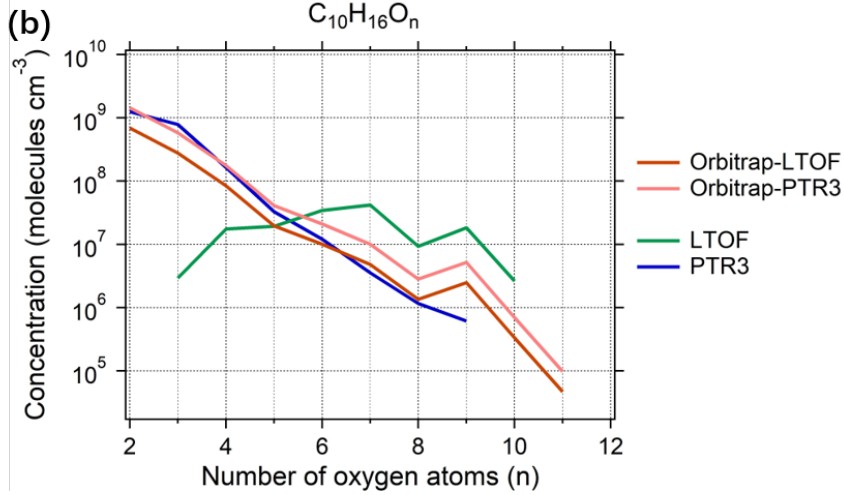

751

**Figure 5:** Estimated concentrations of the main $C_{10}$-monomer oxidation products observed in run 2211.

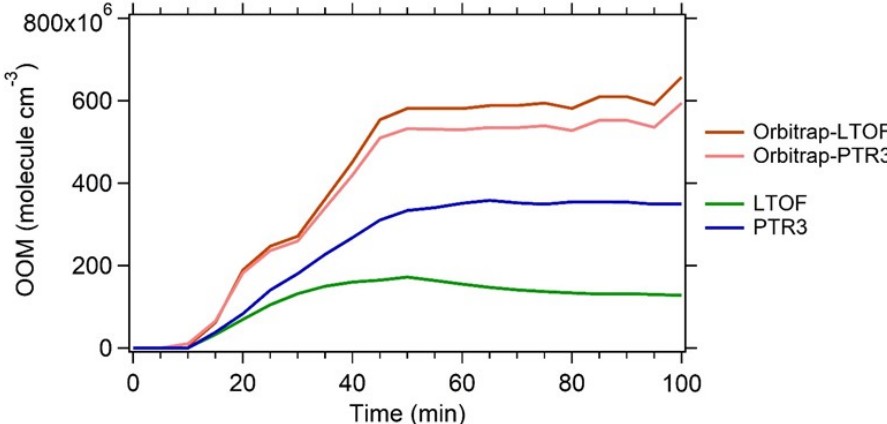

754

**Figure 6:** Estimated concentrations of all measured OOMs in the photooxidation of α-pinene. All monomers $C_{8-10}$ and dimers $C_{18-20}$ measured by CI-($NH_4^+$)-Orbitrap, CI-($NO_3^-$)-LTOF, and PTR3-TOF in run 2213 were summed up. The concentrations of OOM measured by $NH_4^+$-Orbitrap were quantified by the calibration factors derived from correlation analysis between $NH_4^+$-Orbitrap and $NO_3^-$-LTOF (Orbitrap-LTOF, light green) or PTR3-TOF (Orbitrap-PTR3, light blue), respectively.

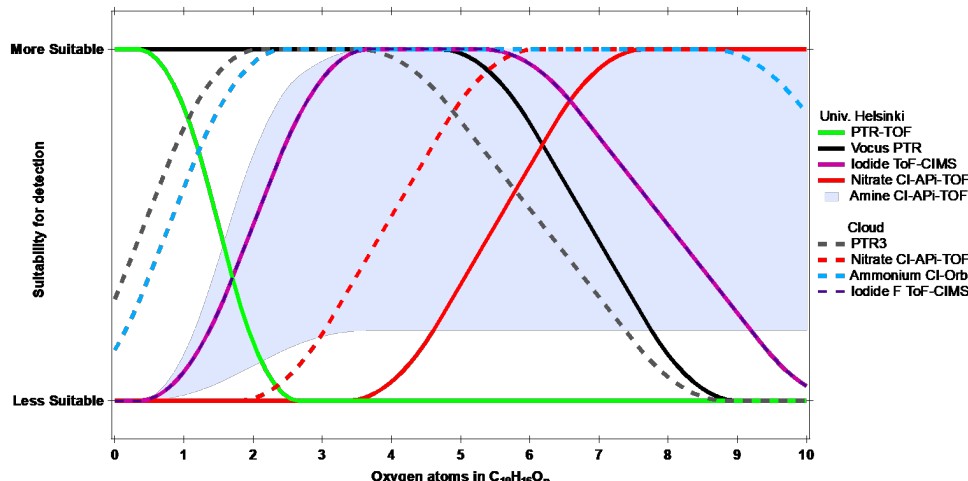

**Figure 7:** Estimated detection suitability of the different CIMS techniques for monomers from α-pinene ozonolysis, plotted as a function of the number of oxygen atoms. Image modified from Riva et al.(Riva et al., 2019b).

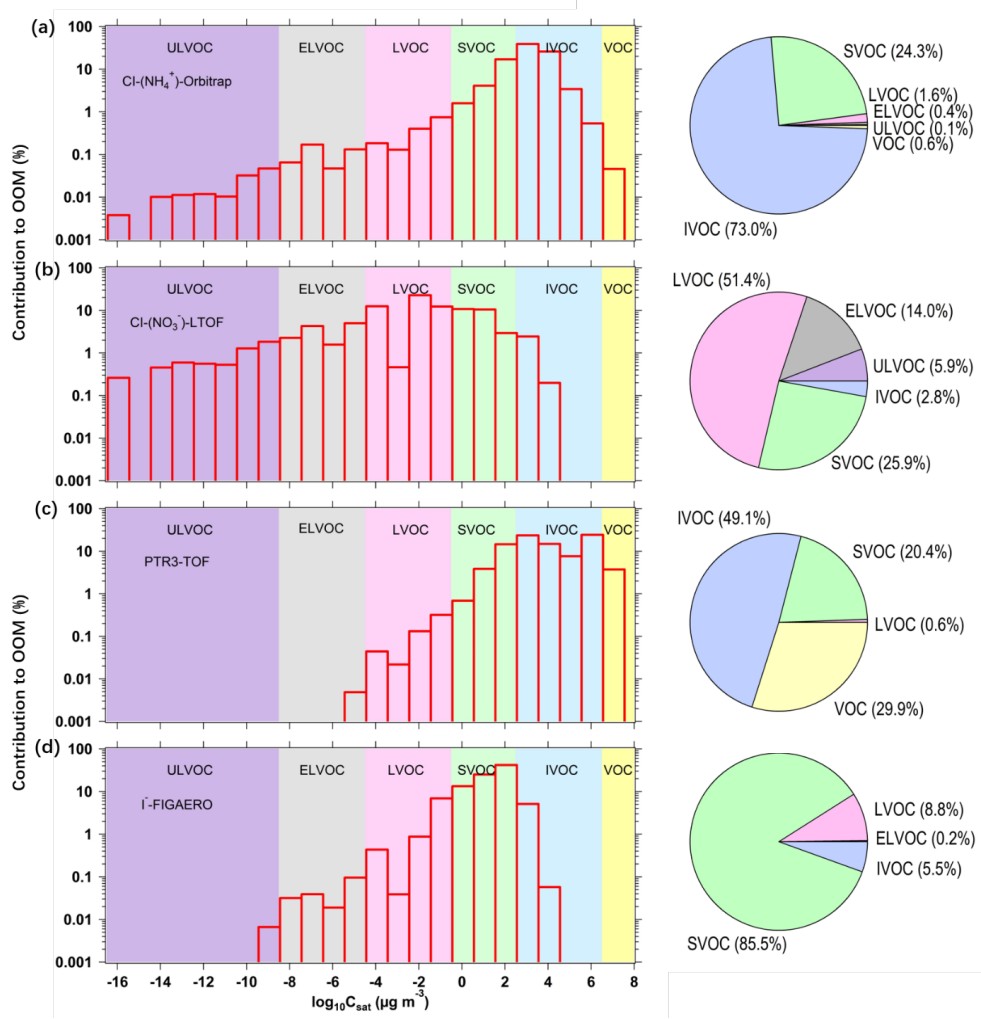

**Figure 8:** Volatility distribution comparison for organic compounds detected by (a) CI-$(NH_4^+)$-Orbitrap, (b) CI-$(NO_3^-)$-LTOF, (c) PTR3-TOF and (d) $I^-$-FIGAERO-CIMS. The background colors represent the saturation concentration ($C_{sat}$) in the range of ultra-low volatility (ULVOCs, purple), extremely low volatility (ELVOCs, gray), low volatility (LVOCs, pink), semi-volatile (SVOCs, green), intermediate volatility (IVOCs, blue) and volatile organic compounds (VOCs). The right pie charts are the corresponding contributions of VOC, IVOC, SVOC, LVOC, ELVOC, and ULVOC classes in run 2211.





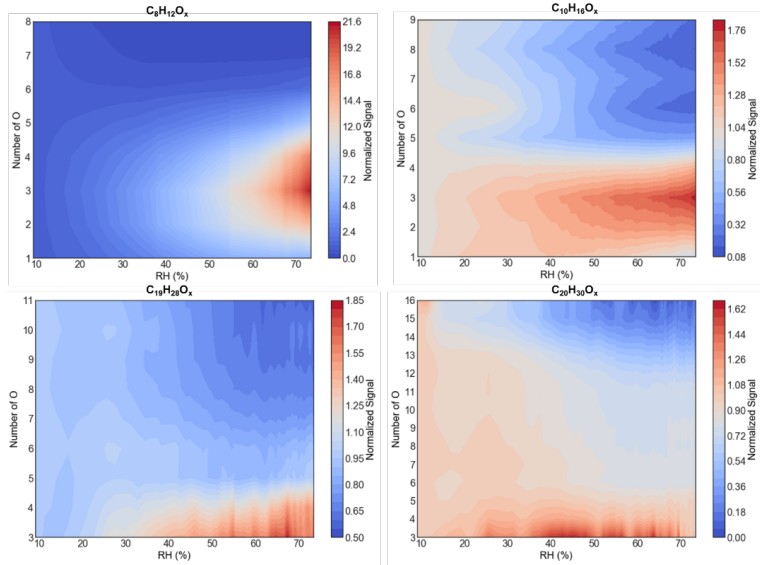

**Figure 9:** The effect of relative humidity on the distribution of the most abundant monomers and

dimers measured by $NH_4^+$-Orbitrap. The RH ramped from ~10% to ~80% in run 2211. The normalized

signal represents the signal variation ratio at certain RH compared to that at RH = 10%, normalized

$signal = \frac{signal_{RH}}{signal_{10\%}}$.