# Peer review of "Ammonium CI-Orbitrap: a tool for characterizing the"

_Atmospheric Measurement Techniques, 2023_

## Author Comment (AC1)

**Referee 1**

The manuscript proposed by Dandan Li et al. entitled "Ammonium *CI-Orbitrap: a tool for characterizing the reactivity of oxygenated organic molecules*" presents the application and potential of a new instruments for the characterisation of a large range of gas phase oxygenated organic molecules (OOMs). OOMs are essential compounds involved in SOA formation and new particle formation processes, their characterization being one of the main challenges in atmospheric chemistry. In that sense, the paper is of great interest for the international scientific community. The paper is clear, and well-structured and contains valuable information. The methodologies regarding the experiments and state of art instruments used are well described, even if some precisions could be added to some extent. The interest of the $NH_4^+$.Orbitrap is evidenced; but the results could be discussed more. As a consequence, I recommend the publication of the paper after the authors address the following points:

-We thank the reviewer for his/her careful consideration of our article. We attached a revised version of the manuscript in which we considered all the comments raised by the reviewer. Below, you will find our point-by-point reply.

**Main comments**

**1 155**: Figure S1 does not support the stability of amines, as it does vary over ca. 1 order of magnitude during the period the period shown on Figure S1. Also I do not understand why a time series of 15 days is presented on Figure S1 while the paper focuses on 2 experiments. It is clear from figure 1 that humidity and to a lesser extent T are affecting amines signal, and it might be discussed in the section 3.6 about RH dependence. The same time series corresponding to runs 2211 and 2213 would be more appropriated than a 15 days time series to support authors statement.

-Thank you for the referee's suggestions. Two runs (2211 and 2213) were chosen to study the performance of the $NH_4^+$-Orbitrap due to the following reasons, (i) α-pinene (AP) ozonolysis is well known compared to other VOCs oxidation processes that were studied within these two weeks. As it is the first study using the $NH_4^+$-Orbitrap coupling, it is more reliable to evaluate the $NH_4^+$-Orbitrap for detecting products from AP+$O_3$ and compare it with previous reports such as Riva et al. (2019). Additionally, $NH_4^+$ mode was applied from run 2209 to 2221 for 15 days. Of these 13 runs, 4 runs aimed to study the nucleation mechanism of $H_2SO_4$-$NH_3$ (runs 2217-2219); 3 runs were for pure isoprene (IP) nucleation (runs 2215-2216) and 2 runs for AP+IP nucleation (runs 2209-2210); and 2 runs with inconstant experiment conditions (runs 2210 and 2221). Therefore, only runs 2211 and 2213 focused AP oxidation under different conditions (RH/$NO_x$) were used to compare the different instruments.

As suggested by the reviewer, we have modified the time series of the amines to focus on the specific runs used in this study. The average signal was $1.9 \times 10^6 \pm 1.2 \times 10^5$ (cps).

[Figure]

**Figure S1** Time evolution of the sum of the 13 amines used to normalize signal intensity in runs 2211 and 2213. Temperature and humidity were also reported throughout the different experiments when the $NH_4^+$-Orbitrap was used.

**2 Experimental approach and product analysis:** A brief comment on how instruments other than NH4+ orbitrap have been calibrated or how quantification estimates were performed is necessary. Even if some well-established methodologies exist. This can be part of Supplementary material if the authors do not want to make the manuscript longer.

- We have added more descriptions regarding the calibration of the $NO_3^-$-LTOF and the PTR3-TOF. The particle phase data of $I^-$-CIMS were not analyzed in this study and the normalized signals of the gas phase were used to compare the volatility range and correlation analysis with the $NH_4^+$-Orbitrap. Hence, the calibration methods were not mentioned in this study.

Lines 185-194: *The $NO_3^-$-LTOF was directly calibrated using sulfuric acid ($H_2SO_4$), where the detection efficiency of HOMs was assumed as similar to $H_2SO_4$ (Kurten et al., 2012). However, OOMs with less oxygen number (O < 6) were prone to a lower detection efficiency compared to $H_2SO_4$, leading to an underestimation (Stolzenburg et al., 2018a; Ehn et al., 2014). A calibration factor C was determined to be ~4.13 × $10^{10}$ molecules $cm^{-3}$ during CLOUD14 (Caudillo et al., 2021). The concentration of OOMs was also corrected using a mass dependent transmission efficiency inferred by depleting the reagent ions with several perfluorinated acids. Assuming that OOMs got lost in sampling lines due to diffusion, the losses of OOMs were corrected with a diffusion coefficient scaling with the molecular mass. More information could be found in former studies (Heinritzi et al., 2016; Stolzenburg et al., 2018; Simon et al., 2020; Caudillo et al., 2021).*

Lines 205-219: *A gas standard mixture containing 1 ppm of 3-hexanone, heptanone, and α-pinene in nitrogen was dynamically diluted by a factor of 1000 in VOC-free air to contain 1 ppbv of each compound, and then was used to calibrate the PTR3-TOF. All data were analyzed using TOF-Tracer software running on Julia 0.6 (https://github.com/lukasfischer83/TOF-Tracer) and were further corrected for the duty cycle transmission of TOF and temperature dependent sampling line losses (Stolzenburg et al., 2018). On the one hand, duty cycle corrected counts per second dcps, $dcps_i = cps_i × (101/m_i)^{1/2}$, was utilized to account for the mass-dependent transmission of the TOF mass spectrometer (Breitenlechner et al., 2017). The calculated sensitivities of 3-hexanone and heptanone were comparable to the observed ones. Therefore, the concentration of oxygenated products was estimated using the sensitivity of 3-hexanone as lower-limit values due to possible fragmentation (Breitenlechner et al., 2017; Stolzenburg et al., 2018). On the other hand, the detected OOMs having (extremely) low volatility were assumed to be lost by diffusion and adjusted by a temperature dependent loss-*

*correction. The sampling line losses considered three loss sections under different temperatures, including losses at the sampling lines within and outside the chamber, and within the PTR3-TOF instrument. Details can be found in previous studies (Breitenlechner et al., 2017; Stolzenburg et al., 2018).*

**2.3** How do the authors differentiate a peak from the background? In online-MS studied; it is commonly assumed that a peak is detected when its area as 3 times higher the standard deviation of noise. Is it what has been done using ORBITOOL?

-The background was determined and removed from the average spectra using Orbitool. The raw data were first averaged to 5 mins, reducing the intensities of noise peaks. Then Orbitool took all detected peaks within a mass range between X+0.5 to X+0.8 Da, where most compounds (i.e., containing C, H, O, S, N) are not located. Signal intensity below a certain percentile is considered as noise, which was set as 70th percentile in this study. Hence, the noise and discriminator levels were calculated as $\mu$ and $\mu + 3\sigma$, where $\mu$ and $\sigma$ were the mean and the standard deviation of the noise signal, respectively. Noise signals lower than the discriminator level were removed from the average spectrum. Details can be found in our previous paper (Cai et al., 2021).

**Section 3.1:** I am not sure this part is necessary, because this is an illustration that an instrument with a high mass resolving power separates more easily isobaric compounds compared to instruments with a lower mass resolution. Any scientist able to understand what a mass resolution of 160 000 compared to 10 000 means is convinced that the first one is far better for separating isobaric compounds (without any demonstration needed). The interest of the paper is not the mass resolution of the orbitrap but its association to $NH_4^+$ as CI. Finally, if the authors find a justification to keep this section, I recommend them to normalize to 1 the Y scale each plot of figure 2.

-There is no doubt that the high mass resolving power of Orbitrap eases the identification of isobaric compounds and its performance of identifying the overlapping peaks has been compared to CI-TOF in our previous studies (Riva et al., 2019a; Riva et al., 2020). An additional aspect to consider is the selectivity of the reagent ion. For example, the $NO_3^-$ ion chemistry is so selective that the higher mass resolving power of the Orbitrap is not critical to resolving the identity of the OOMs. On the contrary, when using reagent ions with very low selectivity, i.e., $NH_4^+$, a greater mass resolving power is necessary to resolve all the ions observed notably those with low signal intensities. Hence, we do believe it is important to highlight the importance of the mass resolving power when using such kinds of reagent ions. Figure 1 is moved to SI considering it is a concept of peak identification and mass resolving power.

**Section 3.2** must be improved based on comments below:

**L.287 288:** Does it make sense to compare $NH_4^+$.orbitrap to another instrument (PTR-3) that is not optimised to compare OOMs? The authors showed the $NH_4^+$.orbitrap is more suited for OOMs detection, but the comparison is not on an equal foot with the PTR-3. Maybe the latter should be excluded from this study?

-We do not mean to conclude that the $NH_4^+$-Orbitrap has a better detection for OOMs compared to the PTR3-TOF. Although the PTR3-TOF was optimized to be sensitive to ammonia, it still

observed many oxygenated species that were used for the correlation analysis and the semi-quantification of the less oxidized OOMs observed by the $NH_4^+$-Orbitrap. This section has been revised as follows:

Line 313-318: *The PTR3-TOF mainly detected compounds below m/Q 300 Th with an average O:C of 0.5 ± 0.3, which was due to the optimization to (i.e., lowering E/N value) measure ammonia and amines sensitively, which ultimately impacted its capability to detect efficiently OOMs. However, many less oxygenated OOMs were still observed by the PTR3-TOF and were used to conduct the correlation analysis of time series with those detected by the $NH_4^+$-Orbitrap.*

**L.288-289**: As mentioned, the quantification limit of the I-.CIMS is higher than gas phase OOMs concentration. Giving a detection limit for each compound detected by each instrument is probably unrealistic, but the information about the range of limit of detection/quantification for each instrument would be helpful for a reader not expert with all these instruments.

While we haven't directly measured the LoD for the instruments, we report LoD from the existing literature.

Line 177: *The limit of detection (LoD) for OOMs is $5 \times 10^4$ molecules $cm^{-3}$ (Simon et al., 2020).*

Line 201-202: *The LoD of PTR3-TOF for detecting OOMs is $8 \times 10^5$ molecules $cm^{-3}$(Breitenlechner et al., 2017).*

Line 225-226: *The LoD of $I^-$-CIMS for OOMs could be lower to $\sim 10^7$ molecules $cm^{-3}$(Lee et al., 2014).*

Line 318-320: *Due to the selectivity and potential losses within the sampling line/inlet of the $I^-$-CIMS equipped with a FIGAERO inlet fewer monomers of $C_{8-10}$ and dimers of $C_{19-20}$ were observed, with an average O:C of 0.5 ± 0.2.*

**L.302-303**: A $R^2 > 0.5$ alone is not a good criterion for "high correlation", as it depends on the number of points associated to each sample, etc. In addition, the good correlation with other instruments could be explained by similar biases, for example. Please temper statements, or strengthen the statistical analysis.

-We agree with the referee that a correlation factor greater $R^2 > 0.5$ alone is not enough to conclude that two parameters are highly correlated. In our case, two runs (run 2211 and 2213) were used to analyze the correlation of time series for the compounds measured by the different instruments. This includes AP injection, steady state stage, $NO_x$ or CO injections, and RH variation. As a result, for one compound, 755 data points were recorded and used for the correlation analysis, providing statistical confidence for the correlation factor. As an example, the time series of $C_{10}H_{14}O_2$ and $C_{10}H_{14}O_{10}$ measured by three instruments in run 2211 are displayed in Figure R1.

[Figure]

Figure R1. Timeseries of $C_{10}H_{14}O_2$ and $C_{10}H_{14}O_{10}$ measured by the $NH_4^+$-Orbitrap (Orbitrap), the $NO_3^-$-LTOF (LTOF), and the PTR3-TOF (PTR3).

More detailed for the semi-quantification method was added as follows:

Line 154-171: *No direct calibration has been performed for the $NH_4^+$-Orbitrap, but a semi-quantitative method was used to estimate the OOMs concentrations based on the correlation analysis using the $NO_3^-$-LTOF or the PTR3-TOF. The values of the Pearson correlation coefficients ($R^2$) were determined between the $NH_4^+$-Orbitrap and two other instruments using the timeseries during two runs (run 2211 and 2213). This includes AP injection, steady state stage, $NO_x$ or CO injections, and RH variation. As a result, for one compound, 755 data points were recorded and used for the correlation analysis. For each instrument (referred to as REF), OOMs with $R^2$ greater than 0.9 (i.e., A) between REF and the $NH_4^+$-Orbitrap, were used to determine a calibration factor ($c_{Orbi-REF}$, molecules $cm^{-3}$) and retrieve the concentrations of OOMs measured by the $NH_4^+$-Orbitrap according to the following equations 4-5:*

$$c_{Orbi-REF} = \frac{[A]_{REF}}{[A]_{nor}} \qquad (4)$$

$$[OOM]_{Orbi-REF} = c_{Orbi-REF} \times [OOM]_{nor} \qquad (5)$$

*The calibration factor between the $NH_4^+$-Orbitrap and REF (~2.62 × $10^8$ for $NO_3^-$-LTOF and ~4.83 × $10^8$ for PTR3-TOF) was assumed to be constant for all the OOMs. However, decomposition of peroxides (i.e., ROOR and ROOH) can be expected within the PTR3-TOF. While fragmentation of dimeric compounds can contribute to the overall signal of the monomers, the concentration of such species remains minor (Li et al., 2022). As a result, we do not expect large enhancement of the monomers signal intensity. Finally, a temperature-dependent sampling-line loss correction factor was applied (Simon et al., 2020).*

The description of the instrumental comparison was modified as follows:

Line 330-339: *The $NH_4^+$-Orbitrap and the $NO_3^-$-LTOF detected OOMs with the same chemical compositions, covering monomers and dimers, among which 18 OOMs showed $R^2 > 0.9$. Regarding the PTR3-TOF, the $NH_4^+$-Orbitrap demonstrated high correlations for most of the monomers and fewer dimers, including 32 species having an $R^2 > 0.9$. Due to potential losses within the FIGAERO inlet, fewer OOMs were detected by the $I^-$-CIMS. However, certain families of compounds, including $C_{10}H_{15}O_{5-7}N$ and $C_{20}H_{31}O_{7,9}N$ showed high correlations (i.e., $R^2 > 0.9$) between the $NH_4^+$-Orbitrap and with the $I^-$-CIMS. Finally, the $NO_3^-$-LTOF was regarded as the reference instrument for HOMs measurements. Only fewer monomers with high oxygen content were detected by the $NO_3^-$-LTOF and the PTR3-TOF, and only a few dimers between the $NO_3^-$-LTOF and the $I^-$-CIMS with moderate relevance.*

**L.353**: It is not clear if the raw signal (i.e., counts) or concentrations have been used here? If raw signals are used, can the authors justify their choice? And would Figure 8 be different if the concentrations are used instead of signals?

-Concentrations were used for the $NO_3^-$-LTOF and the PTR3-TOF, while the signal intensities were used for the $NH_4^+$-Orbitrap and the $I^-$-CIMS. It is important to point out that using concentration or raw signal will not change the relative volatility distribution for a given reagent ion as concentrations are determined by applying a unique calibration factor (e.g., $4.13 \times 10^{10}$ molecules cm$^{-3}$ for the $NO_3^-$-LTOF).

**Section 3.4**: a simple comparison on couple of common compounds detected by both PTR-3 and NO3-CIMS would be nice to validate their quantification, showing there. Both instruments are used as reference to "calibrate" $NH_4^+$.orbitrap, but are PTR-3 and NO3-CIMS consistent when measuring the same compound? In addition, the $NH_4^+$.orbitrap falls in a factor 2 comparing with other instruments, which is satisfying and reasonable considering all the uncertainties associated with quantification on online-MS, but cannot be qualified as "good", which is subjective term.

-There were some species like $C_8H_{12}O_4$ which was consistent between the $NO_3^-$-LTOF and the PTR3-TOF as depicted in Figure R2.

[Figure]

Figure R2. The timeseries of $C_8H_{12}O_4$ measured by the $NH_4^+$-Orbitrap (Orbitrap), the $NO_3^-$-LTOF (LTOF), and the PTR3-TOF (PTR3).

While the two instruments are not necessarily consistent for an extensive range of compounds, such as $C_{10}H_{14}O_2$ or $C_{10}H_{14}O_{10}$ (Figure R1); The $NO_3^-$-LTOF and the PTR3-TOF cannot be directly compared as they measure different types of OOMs due to the selectivity of the reagent ions. The former was sensitive to HOMs with $n_O > 6$ while the latter detected efficiently OOMs with $n_O = 1\sim5$. We revised the description of the $NH_4^+$ correlation results as follows:

Lines 347-353: *The concentrations of $C_{10}$-monomers measured by the $NH_4^+$-Orbitrap based on the two calibration factors vary within a factor of 2, which indicates the consistency between the two correlation analyses. The variation trend of concentrations with the oxygen number of the $NH_4^+$-Orbitrap is similar to that of the $NO_3^-$-LTOF in the range of $n_O>6$, and it is similar to that of the PTR3-TOF in the range of $n_O=1\sim5$. Taking into consideration that such ranges are also the oxygen number ranges with high sensitivities respectively, this proves the robustness of the the $NH_4^+$-Orbitrap and the semi-quantification method.*

**Section 3.6**: the discussion is interesting here, but the results should be more detailed. For example, it is not discussed that intensity of $C_8$ compounds increased whatever the number of O atoms, while it is more contrasted for other compounds. In addition, this increase can be up to ca. 20 for $C_8H_{12}O_{2-4}$ compounds, while it is limited to 1.6 for $C_{10}$, $C_{19}$ and $C_{20}$. Is there an explanation here? In addition, based on Figure S6, it seems the effect of RH is very important at $n_O<8$, but what about the effect of $n_C$? As I just mentioned, the effect of RH seemed to be stronger for $C_8$ compounds. The increased in polarity or O/C with decreasing C number might be an explanation? This must be investigated. Figure S6 also evidenced that I-.FIGAERO-CIMS sensitivity is only decreasing with increasing RH, while it is not the case for other instruments. The authors should comment this result. Finally, as the authors cannot distinguish the effect of increasing RH on chemical and physical processes (based on experiments presented in the present paper), it is evident that RH influences $NH_4^+$.Orbitrap sensitivity, that can be different for each OOM, but this specific effect requires more attention and dedicated studies before the $NH_4^+$.Orbitrap can be used in field studies (for example, injection of pure or

mixture of standards in atmospheric chamber at varying RH). From what is presented here, the understanding of RH effect on the $NH_4^+$.Orbitrap capabilities is too scarce to be able to understand the time series evolution of OOMs that would be obtained in the real atmosphere.

-We do not want to speculate on the evolution of the $C_8$ compounds and other compounds with shorter carbon skeletons as such species are suspected to be formed from heterogeneous/wall reactions of $C_9$-$C_{10}$ compounds produced from the gas phase oxidation of monoterpene. Figure S7 has been reported as a reference to underline the potential bias when measuring oxidation products when changing humidity. Instead, we focused on monomers and dimers that are produced from well-known gas phase reactions. In addition, the scope of the paper is to investigate $NH_4^+$ ion-based chemistry not to investigate the RH effect of chemical ionization at medium and atmospheric pressure. We do agree with the reviewer that a dedicated study should focus on the RH effect to measure OOMs in the atmosphere using different ion chemistry.

**Minor comments:**

**Section 3.1:** OVOCs should be replaced by OOMs, as most of detected compounds are not volatiles.

-Revised.

Figure 5: caption should be more explicit.

-We add more description about current Figure 4 as follows:

***Figure 4:*** *Estimated concentrations of the main $C_{10}$ oxidation products (a) $C_{10}H_{14}O_n$ and (b) $C_{10}H_{16}O_n$ as a function of oxygen numbers observed in run 2211. Orbitrap-LTOF and Orbitrap-PTR3 represented the estimated concentration of monomers measured by $NH_4^+$-Orbitrap using the calibration factors from the correlation analysis with $NO_3^-$-LTOF and PTR3-TOF, respectively.*

Figure 7: here the PTR-3 is probably limited compared its real potential to OOMs, because it has been tuned for efficient detection of NH4+…

1.  409: is.

-Revised.

2.  757: concentrations of OOMs

-Revised.

---

## Author Comment (AC2)

Referee 2

This work studied CI-NH4-Orbitrap as a powerful tool for characterizing oxygenated organic molecules (OOMs) from atmospheric oxidation of VOCs. The manuscript compares the performance of CI-NH4-Orbitrap with a few other chemical ionization based mass spectrometers with a range of ionization methods and resolving power. The comparison showed that CI-NH4-Orbitrap is a promising instrument to more comprehensively characterize and even quantify a near-complete range of OOMs from oxidation. The work is solid and well written. It will likely deserve publication at AMT. But some sections of the manuscript need to be better clarified and some in-depth discussion is needed.

-We thank the reviewer for his/her careful consideration of our article. We attached a revised version of the manuscript in which we considered all the comments raised by the reviewer. Below, you will find our point-by-point reply.

Detailed comments:

1. Line 39 in Abstract. Change "highly oxidized volatile organic compounds (HOM)" to "highly oxidized molecules (HOM)".

 -Revised to "highly oxidized organic molecules" based on Bianchi et al., 2019.

2. Line 56-58. OOMs can also be generated through bimolecular RO2 pathways not involving autoxidation. Autoxidation is important, but review OOM formation more compressively, other pathways should also be mentioned here.

 -We revised the statement as follows:

Line 58-60: *OOMs can be generated through the bimolecular peroxy radicals ($RO_2$) pathway or by the autoxidation of $RO_2$ followed by the termination pathways (Bianchi et al., 2019; Mohr et al., 2019)*

3. Line 76-87. In the negative ion-based MS, it would be helpful to also mention iodide-CIMS, as it is compared later in the text. One sentence to set up the context would be a good idea, also because iodide-CIMS measures a wide range of OOMs.

 -We add a sentence to describe iodide-CIMS in the main text:

Line 75-78: *For example, negative ion-based chemistry, including nitrate ($NO_3^-$), can optimally detect HOMs, which only constitute a small subset of the OOMs (Lee et al., 2014; Berndt et al., 2018; Riva et al., 2019b); iodide ($I^-$) can efficiently detect various OOMs with 3-5 oxygen atoms (Riva et al., 2019b; Lee et al., 2014).*

4. Line 150. Change the sentence to "The NH4+ reagent ion cannot be directly detected due to…"

-Revised.

5. Line 156. In prior studies using the same NH4+ ionization but with a Tofwerk LTOF, is there evidence regarding the ratio of the sum of these "surrogates" over the reagent ion abundant? If this data is available, it would be useful to mention here.

-A direct comparison of the cluster and the presence of amines is not possible. First, the Orbitrap cannot detect masses lower than 50 Th preventing the detection of the reagent ions. Secondly, in Tofwerk LTOF (i.e., Vocus-$NH_4^+$) the distribution of the reagent ions, i.e., cluster distribution is highly dependent on the user settings (i.e., pressure, $NH_3$ flow, Vocus RF and voltage settings) and the ion transmission related to the instruments. Finally, no Vocus-$NH_4^+$ instrument was deployed during the Cloud campaign. As a result, it is not possible to perform such kind of comparison.

6. Line 159-166. The uncertainties regarding this semi-quantification method needs to be discussed somewhere. For example, this method assumes that the C10H14,16Ox formulas measured by the three instruments are the same species without artifacts? Do dimers in PTR3 decompose to monomers? How can you obtain the calibration factor from correlation analysis alone (cps vs. cps between different instruments)? Do you need response factors (e.g., ppt/cps) from either the PTR3 or NO3-LTOF to get concentration results for Orbitrap?

By the way, NO3-LTOF is termed in this paragraph, but termed "CI-NO3-LTOF" or "CI-NO3-APi-LTOF" in other places. The terminology needs to be consistent throughout the manuscript.

-Decomposition of peroxide can be expected as it has been shown for the Vocus PTR, however as discussed by Li et al., (2022) protonation of peroxides (i.e., ROOR and ROOH) can partly lead to the decomposition of the analytes. Determining to which extent the ionization process within the PTR3 fragments peroxide is beyond the scope of this study as it is by itself a dedicated work (e.g., Li et al., 2022). While fragmentation of dimeric compounds can contribute to the overall signal of the monomers as mentioned by Li et al., 2022, the concentration of such species remains minor (as shown in many previous studies and within this work). As a result, we do not expect large enhancement of the monomers signal intensity.

A sentence has been added to mention this potential artifact:

Lines 154-156: *No direct calibration has been performed for the $NH_4^+$-Orbitrap, but a semi-quantitative method was used to estimate the concentrations of OOMs measured by $NH_4^+$-Orbitrap based on the correlation with $NO_3^-$-LTOF or PTR3-TOF.*

Lines 166-171: *However, decomposition of peroxides (i.e., ROOR and ROOH) can be expected within the PTR3-TOF. While fragmentation of dimeric compounds can contribute to the overall signal of the monomers, the concentration of such species remains minor (Li et al., 2022). As a result, we do not expect large enhancement of the monomers signal intensity. Finally, a temperature-dependent sampling-line loss correction factor was applied (Simon et al., 2020)*

We also uniformed the terms of $NH_4^+$-Orbitrap, $NO_3^-$-LTOF, PTR3-TOF and $I^-$-CIMS throughout the manuscript.

7. Line 258-267. The dm threshold is also dependent on the level of knowledge of the possible chemical formulas at the normal m/z. In case the chemical formulas are known at high confidence (in the example of known VOC precursors), the threshold may be smaller. But in real cases where more than two peaks are present, the threshold can be larger. This is a very complex issue. The simplified illustration here is certainly useful, but some more in-depth discussion is warranted in real cases.

-We revised this part as follows:

Lines 301-305: *It should be noted that the $NH_4^+$-Orbitrap has shown its strength in separating neighboring peaks in controlled experiments, in which the knowledge of the chemical compositions for OOMs is relatively abundant. The advantages of higher mass resolving power should be further stressed in ambient observations, where the knowledge about OOM species can be limited with a larger number of detectable peaks.*

8. Figure 1. The last plot was not described in the caption.

-We moved Figure 1 to SI considering it is a concept of peak identification and mass resolution. The description of the last plot was added in the caption as follows:

***Figure S3*** *Simulated TOF spectra of overlapping peaks of different intensities near m/z 200 (a, b, c), and the ratio of dm to FWHM as a function of peak height ratio (d). Assuming a TOF mass analyzer with a mass resolving power of 8,000, somewhere between a Tofwerk HTOF ("high-resolution time-of-flight') and LTOF ("long high-resolution time-of-flight) mass spectrometers. FWHM was the full width at half maximum and dm was the distance between two overlapping peaks. Peak height ratio represented the signal intensity ratio of overlapping peaks and peak distance referred to the ratio of dm to FWHM. The overlapping area represented a greater proportion of the peak area of the less intense peak. The noise wasn't added to the data.*

9. Line 280. In the comparison between CI-NH4-Orbitrap and I-FIGAERO-CIMS, it is unclear that the large difference is number is mainly due to the less selectivity of NH4+ ionization or the higher resolving power of the Orbitrap. Some clarification is needed. The range of oxygen number seems comparable based on previous studies of iodide-CIMS (from nO=2 to HOMs). The detection limit issue mentioned in Line 289 seems to suggest that this difference is largely due to instrument sensitivity tuning issue for iodide-CIMS? If this is the case, the comparison does not really speak for the advantages of CI-NH4-Orbitrap in ionization method and resolving power.

-No, we did not mean to mention that the $I^-$-CIMS equipped with a FIGAERO inlet has a higher detection limit for OOMs. Iodide showed a higher selectivity to OOMs, which detected sufficiently the semi-volatility OOMs with 3-5 oxygen atoms but less the most oxidized OOMs which might arise from losses within the sampling line and the inlet as the instrument was optimized to collect/analyse aerosol particles. We revised the statements as follows:

Line 318-320: *Due to the selectivity and potential losses within the sampling line/inlet of the $I^-$-CIMS equipped with a FIGAERO inlet fewer monomers of $C_{8-10}$ and dimers of $C_{19-20}$ were observed, with an average O:C of 0.5 ± 0.2.*

10. Line 284. How was the O/C ratio estimated? 0.4+/- 0.2 seems to be a very large uncertainty. Is this due to the variation between the two experiments? Or uncertainties in the semi-quantification method? With the accurate formula detection, I would expect smaller uncertainties in O/C ratios.

-The ratio of O to C was calculated based on the assigned formula by each mass spectrometer. Only Run 2211 was used to plot the mass defect figure and the marker size was scaled to signal intensities for $NH_4^+$-Orbitrap. 0.2 indicated the variation of O/C, not the uncertainty of the molecular formula identified by the $NH_4^+$-Orbitrap. The $NH_4^+$-Orbitrap identified ~460 OOMs and the O/C varied from 0.06 to 1.1, the corresponding formulae were $C_{10}H_{10}O$ and $C_8H_{14}O_9$.

11. Line 294. This number suggests a large fraction of the chemical formulas detected by CI-NH4-Orbitrap are not seen by any of the other three instruments. What are the characteristics (e.g., number of C, H, O, O/C, etc.) of the chemical formulas co-detected vs. only detected by CI-NH4-Orbitrap? Combining PTR3-TOF, Iodide-CIMS, and NO3-LTOF, it appears to me that the overall selectivity is comparable to CI-NH4-Orbitrap. If the difference boils down to the discrepancies regarding sensitivity (e.g., Orbitrap much better than the others and most of the formulas only detected by CI-NH4-Orbitrap are relatively small peaks) and resolving power (e.g., CI-NH4-Orbitrap detects formulas at high confidence, but the other instruments do not), I think it is worth discussing this difference to highlight the superior performance of CI-NH4-Orbitrap.

 -After we re-checked the dataset, we found a mistake. The actual number is ~42%. Generally, OOMs co-detected by the different chemical ionization techniques show clear characteristics in oxygen number: OOMs with an oxygen number greater than 4 are co-detected by the $NO_3^-$-LTOF and the $NH_4^+$-Orbitrap, while those co-detected by the PTR3-TOF and $NH_4^+$-Orbitrap have an oxygen number < 7 (Figure R4).

For the overall selectivity, we must argue that although there are molecules only detected by the $NH_4^+$-Orbitrap, there are still other molecules (e.g., $C_{18}H_{30}O_{10}$ in $NO_3^-$-LTOF, $C_4H_8O_2$ in the PTR3-TOF) that could only be detected by the $NO_3^-$-LTOF or the PTR3-TOF. Considering that the PTR3-TOF has been optimized for measuring ammonia and amines in this study and is not in the best state for measuring a wider range of OOMs, we cannot conclude that the $NH_4^+$-Orbitrap is better than other mass spectrometers in measuring all the OOMs. It should be stressed out that the $NH_4^+$-Orbitrap can measure the widest range of oxygen numbers, as shown in Figure R4.

Lines:322-324: *Due to differences in selectivity and sensitivity of the analytical methods toward OOMs, ~42% of the identified species by $NH_4^+$-Orbitrap are simultaneously detected by other mass spectrometers.*

We add Figure R4 to SI as Figure S4.

[Figure]

*Figure S4* *The fractions of co-detected OOMs with other instruments among those detected by NH$_4^+$-Orbitrap with the variation of oxygen number. Purple areas represent the OOMs only detected by NH$_4^+$-Orbitrap, which account for approximately 42%; yellow areas were OOMs co-detected by NH$_4^+$-Orbitrap and NO$_3^-$-LTOF; red areas were OOMs co-detected by NH$_4^+$-Orbitrap and PTR3-TOF; and blue areas were OOMs co-detected by the three mass spectrometers.*

12. Figure 3. It would be helpful to draw a few lines indicating the major chemical formula series detected by the different instruments in the KMD plots.

-We changed the mass defect with a few lines indicating the major families of OOMs as follows:

[Figure]

*Figure 2:* *Mass defect plots for organic compounds measured by (a) NH$_4^+$-Orbitrap, (b) NO$_3^-$-LTOF, (c) PTR3-TOF and (d) I$^-$-CIMS in run 2211. The x-axis represents the mass-to-charge ratio of the neutral analyte and the y-axis represents the corresponding mass defect, which is the difference between their exact mass and nominal mass (Schobesberger et al., 2013). Markers were all sized by the logarithm of their corresponding signals and colored by the O:C value. Some major OOMs measured by different instruments were indicated by the black lines.*

13. Section 3.3. It is useful to describe a few major chemical formulas which have the worst correlations. Although it is not a chemistry paper, but providing such information can help others think about the chemical reasons behind these correlations.

- There are no clear elemental characteristics among the worst correlated species, as shown in Table R1. When checking the time series of the worst correlated OOMs, we found there were complex reasons why the correlations were lower. For C$_{18}$H$_{32}$O$_3$ detected by the NH$_4^+$-Orbitrap and the PTR3-TOF or C$_{18}$H$_{30}$O$_7$ detected by the NH$_4^+$-Orbitrap and the NO$_3^-$-LTOF, although

the variation trend was similar during part of the experiment, there were clear differences during other periods. This might be because given species were too close to the LoD of one instrument yielding larger uncertainties (Figure R5 and R6). We cannot also rule out that different experimental conditions would have led to the formation of isomers having different sensitivities toward the reagent ions used in this study.

Table R1. The element composition of 20 worst correlation OOMs co-detected by $NH_4^+$-Orbitrap and other instruments

| Item | $NH_4^+$-Orbitrap & PTR3-TOF | | | | $NH_4^+$-Orbitrap & $NO_3^-$-LTOF | | | |
|------|----|----|---|---|----|----|---|----|
|      | C  | H  | N | O | C  | H  | N | O  |
| 1    | 18 | 32 | 0 | 3 | 12 | 16 | 0 | 6  |
| 2    | 10 | 12 | 0 | 4 | 15 | 24 | 0 | 15 |
| 3    | 8  | 12 | 0 | 6 | 20 | 32 | 0 | 20 |
| 4    | 3  | 6  | 0 | 1 | 20 | 31 | 1 | 7  |
| 5    | 10 | 16 | 0 | 7 | 6  | 12 | 0 | 5  |
| 6    | 10 | 14 | 0 | 8 | 18 | 30 | 0 | 7  |
| 7    | 9  | 14 | 0 | 6 | 8  | 11 | 1 | 6  |
| 8    | 6  | 11 | 1 | 1 | 9  | 13 | 1 | 4  |
| 9    | 10 | 20 | 0 | 2 | 12 | 18 | 0 | 8  |
| 10   | 7  | 13 | 1 | 1 | 12 | 22 | 0 | 5  |
| 11   | 10 | 16 | 0 | 8 | 12 | 20 | 0 | 9  |
| 12   | 10 | 18 | 0 | 7 | 19 | 29 | 1 | 6  |
| 13   | 10 | 17 | 0 | 4 | 19 | 30 | 0 | 10 |
| 14   | 8  | 14 | 0 | 7 | 7  | 10 | 0 | 7  |
| 15   | 4  | 6  | 0 | 2 | 7  | 11 | 1 | 5  |
| 16   | 6  | 9  | 1 | 2 | 8  | 12 | 0 | 7  |
| 17   | 10 | 14 | 0 | 6 | 4  | 6  | 0 | 2  |
| 18   | 4  | 6  | 0 | 1 | 18 | 26 | 0 | 5  |
| 19   | 10 | 20 | 0 | 3 | 5  | 10 | 0 | 3  |
| 20   | 4  | 6  | 0 | 3 | 8  | 13 | 0 | 8  |

[Figure]

Figure R5. The timeseries of $C_{18}H_{32}O_3$ measured by $NH_4^+$-Orbitrap and PTR3-TOF.

[Figure]

Figure R6. The timeseries of $C_{18}H_{30}O_7$ measured by the $NH_4^+$-Orbitrap and the $NO_3^-$-LTOF.

14. Figure 5. The figure legend needs to be explained in figure caption. LTOF means NO3-LTOF? Orbitrap-LTOF means Orbitrap-derived concentrations using NO3-LTOF calibration factors? Does the NO3-LTOF actually measure oxygen number down to 2? Do the concentrations in Orbitrap depend on quantification by PTR3 (proton transfer kinetics) or NO3-LTOF (H2SO4 as the sole standard)? If so, they need to be mentioned. I believe that much of the unclarity stems from Eq. (3), as mentioned in my above comment #6. What are the units of c and [X]? If c is unitless (i.e., cps/cps from correlation analysis), and [X] is in concentration (ppt or molecules cm-3), the equation does not make sense because the second term in the right side of the equation is unitless.

-The answers should be "Yes" for the first four questions. We agree some descriptions should be added to clarify the meaning of Figure 5. The unit of [OOM] is cps (signals detected by $NH_4^+$-Orbitrap) and that of C is molecules cm$^{-3}$, which is the slope of the correlation analysis between the measured concentrations (molecules cm$^{-3}$) from reference instruments and the normalized signal (unitless). We add more description regarding Figure 4, the quantification of the $NH_4^+$-Orbitrap, and the units in Eq. (3-5) in Section 2.2 as follows:

*Figure 4: Estimated concentrations of the main $C_{10}$ oxidation products (a) $C_{10}H_{14}O_n$ and (b) $C_{10}H_{16}O_n$ as a function of oxygen numbers observed in run 2211. Orbitrap-LTOF and Orbitrap-PTR3 represented the estimated concentration of monomers measured by the $NH_4^+$-Orbitrap using the calibration factors from correlation analysis with the $NO_3^-$-LTOF and the PTR3-TOF, respectively.*

Lines 153-171:

$$[OOM]_{nor} = \frac{[(OOM)-NH_4^+] + [(OOM-H)^+]}{\sum[Amine]} \tag{3}$$

*No direct calibration has been performed for the $NH_4^+$-Orbitrap, but a semi-quantitative method was used to estimate the concentrations of OOMs based on the correlation with the $NO_3^-$-LTOF or the PTR3-TOF. The values of the Pearson correlation coefficients ($R^2$) were determined between the $NH_4^+$-Orbitrap and two other instruments using the timeseries during two runs (run 2211 and 2213). This includes AP injection, steady state stage, $NO_x$ or CO*

*injections, and RH variation. As a result, for one compound, 755 data points were recorded and used for the correlation analysis. For each instrument (referred to as REF), OOMs with $R^2$ greater than 0.9 (i.e., A) between REF and the $NH_4^+$-Orbitrap, were used to determine a calibration factor ($c_{Orbi-REF}$, molecules cm$^{-3}$) and retrieve the concentrations of OOMs measured by the $NH_4^+$-Orbitrap according to the following equations 4-5:*

$$c_{Orbi-REF} = \frac{[A]_{REF}}{[A]_{nor}} \tag{4}$$

$$[OOM]_{Orbi-REF} = c_{Orbi-REF} \times [OOM]_{nor} \tag{5}$$

*The calibration factor between the $NH_4^+$-Orbitrap and REF (~2.62 × 10$^8$ for $NO_3^-$-LTOF and 4.83 × 10$^8$ for PTR3) was assumed to be constant for all the OOMs. However, decomposition of peroxides (i.e., ROOR and ROOH) can be expected within the PTR3-TOF. While fragmentation of dimeric compounds can contribute to the overall signal of the monomers, the concentration of such species remains minor (Li et al., 2022). As a result, we do not expect large enhancement of the monomers signal intensity. Finally, a temperature-dependent sampling-line loss correction factor was applied (Simon et al., 2020).*

15. Line 327 and Figure 6. What are the fractions of the reacted carbon measured by these instruments? Table S1 does not show the steady-state a-pinene concentrations, so it is not possible to estimate these fractions by audience. It is also unclear how the remaining formulas only detected by Orbitrap is treated here. Are they also quantified by the same calibration factors?

-We agree that if the "fractions of reacted carbon" was presented as a parameter, it would be better to quantify the true value of the fractions. However, as the carbon closure is not a key point in this research, we revised the paragraph and focused the discussions on the lower selectivity of the $NH_4^+$-Orbitrap, compared to the $NO_3^-$-LTOF and the I$^-$-CIMS. We clarified the steady-state concentration of α-pinene in Table S1 and the time series of the precursors are shown in Figure S5 for both experiments. We revised the paragraph as follows:

Lines 363-368: *The concentrations of OOMs measured by the $NH_4^+$-Orbitrap were higher than both the $NO_3^-$-LTOF and the PTR3-TOF which was optimized for measuring ammonia and amines. This indicates that the $NH_4^+$-Orbitrap can provide a better constraint on the concentrations of the primary products. As an example, pinonaldehyde (i.e., $C_{10}H_{16}O_2$), as one of the most abundant oxidation products, was not efficiently detected by $NO_3^-$-LTOF, which is consistent with the higher selectivity of the $NO_3^-$ reagent ion.*

16. Line 372-376. Should this be due to increased partitioning of water-soluble compounds to the aerosol liquid water? If increased RH leads to partitioning of SVOCs, why did nO<5 signals increase? Presumably these C8H12O<5 species are SVOC and with the enhanced partitioning, their signals in the gas phase should decrease. The explanation is in contrary to the observation. The changing ionization efficiency and multiphase chemistry described in the following paragraph could be the main reasons. I suggest revising this section. The way it is written is confusing.
-The analysis of the RH effect on SOA component partitioning has been discussed in our collaborative study (Surdu et al., 2023). However, in this study, we do not want to speculate on the evolution of the $C_8$ compounds and other compounds with shorter carbon skeletons. Such species are suspected to be formed from heterogeneous/wall reactions of $C_9$-$C_{10}$ compounds,

so RH influences not only their partitioning but also potentially their formation and sinks. In addition, the scope of the paper is to investigate $NH_4^+$ ion-based chemistry not to investigate the RH effect of chemical ionization at medium and atmospheric pressure. We do agree with the reviewer that a dedicated study should focus on the RH effect to measure OOMs in the atmosphere using different ion chemistry. We also agree that the previous version might lead to confusion for the readers, so we rearrange the logic in section 3.6 as follows:

Line 406-449: *The sensitivity of the reagent-adduct ionization has been reported to be affected by the presence of water vapor for a variety of reagent ions (Lee et al., 2014; Breitenlechner et al., 2017). The impact of RH on the detection of OOMs by the $NH_4^+$-Orbitrap was also studied. While the concentrations of gas phase precursor and oxidant remained constant, the RH was raised from 10% to 80%. During this increase the signal of organic vapor behaved inconsistently under an otherwise constant gas-phase production rate (Surdu et al., 2023) and an increase in the condensation sink (Fig. S5). As shown in Fig. 9, the $NH_4^+$-Orbitrap demonstrated an RH dependence. For instance, the signal of less oxygenated molecules (i.e., $n_O < 5$) increased with increasing RH, especially compounds with $n_C = 8$; while the signal of highly oxygenated molecules (i.e., $n_O > 10$) decreased as a function of RH. The average behavior of all $C_{8-10}$ monomers and $C_{18-20}$ dimers was summarized and compared between four instruments (Fig. S6). The other three mass spectrometers also showed obvious RH dependence. Similar to $NH_4^+$-Orbitrap, OOMs with $n_O < 5$ measured by $NO_3^-$-LTOF and PTR3-TOF increased at high RH, and a reverse tendency for HOMs with $n_O > 11$, while OOMs with $n_O = 8\sim11$ seemed to be independent to RH. The large variations of OOMs intensity at different RH measured by $NH_4^+$-Orbitrap may be due to the widest range of oxygen atoms. The causes why OOMs with different oxygen numbers measured by four instruments changed with RH was not clear. Here, multiple possible reasons were provided to explain the signal evolution of the ions with changing RH, such as water affecting the ionization efficiency or altering the physicochemical processes of the gas phase chemistry.*

*First, the efficiency of a particular compound partly relied on whether water vapor competes with the ammonium ion, lowering the sensitivity, or whether it acted as a third body to stabilize the ammonium-organic analyte cluster by removing extra energy from the collision, raising the sensitivity (Lee et al., 2014). $NH_4^+$ primary ions can cluster with water molecules when humidity increased, thereby reducing the clustering of the $NH_4^+$ with organic analytes (Breitenlechner et al., 2017). However, the formed $NH_4^+X_n$ (X being $NH_3$ or $H_2O$; n = 1,2) clusters might also act as reagent ions and ionize OOMs through ligand switching reactions, which were expected to be fast and thus improve the charging efficiency (Hansel et al., 2018). Compared to previous $NH_4^+$-CIMS, the $NH_4^+X_n$ reagent ions were expected to be larger due to the absence of the field in the ion-molecular-reaction zone in Orbitrap, resulting in greater ligand exchanging and increasing the sensitivity for the less oxygenated species (Canaval et al., 2019).*

*For RH-independent compounds, this may be due to the existence of very stable complexes with $NH_4^+$ reagent ion, or sufficient internal vibrational modes to disperse extra energy from the collision (Lee et al., 2014). The highly oxygenated dimers in the category of ULVOCs and ELVOCs which largely partition to the particle phase regardless of the presence of water might indicate that water may also affect the physicochemical processes (i.e., multiphase chemistry, partitioning, etc.), in this case possibly leading to an increase in the driving force of gas-particle partitioning of highly oxygenated species (Surdu et al., 2023), and/or causing the decomposition of highly oxygenated molecules in the particle phase to create less and moderately oxygenated products, e.g., $C_8H_{12}O_{1-5}$ (up to a 30-fold increase in the gas phase) (Pospisilova et al., 2020), although which in the range of SVOCs (e.g., $C_8H_{12}O_{4,5}$) was also*

*thought to partition more to the particle phase at higher RH (Surdu et al., 2023). Finally, while water vapor could affect the gas-phase chemistry through water reactions with the Criegee intermediates (CIs), $HO_2$ chemistry, OH radical concentration, no clear evidence has been identified as earlier discussed by Surdu et al (2023). However, the accurate reasons needs to be further verified in target control experiments like changing the RH in IMR of CI inlet.*

---

## Author Response (AR2)

The black color of the text in this document shows the reviewer comments, while green color shows the authors' responses and the revised text is shown in italics.

In the revision, the authors have addressed most of my prior comments. But there are a few remaining ones:

-We thank the reviewer for his/her careful consideration of our article. We attached a revised version of the manuscript in which we considered all the comments raised by the reviewer. Below, you will find our point-by-point reply.

1. My initial comment #6. The first question to suggest adding uncertainty discussion was not replied or addressed. The method assumes all the C10H14,16Ox formulas measured by the three instruments are the species. Is this valid?

Response: We added an uncertainty discussion in the revised text, and further revise the manuscript to clarify our statements in this version. Instead of all the $C_{10}H_{14,16}O_x$ formulas, we only assumed those with Pearson correlation coefficients ($R^2$) of time series greater than 0.9 between $NH_4^+$-Orbitrap and another instrument (18 species for $NO_3^-$-LTOF and 32 species for PTR3-TOF) were produced from the same chemical process and yielding likely the same species. The correlation factor was derived based on the average sensitivities of these species. For other compounds, we believe they could represent different isomers or arise from decomposition/fragmentation of larger species.

*Line 154-172: Correlation analysis were performed between the $NH_4^+$-Orbitrap and the two reference instruments including $NO_3^-$-LTOF and PTR3-TOF (referred to as REF). The Pearson correlation coefficients ($R^2$) were determined using the timeseries during two runs (run 2211 and 2213). This included AP injection, steady state stage, $NO_x$ or CO injections, and RH variation. As a result, for one compound, 755 data points were recorded and used for the correlation analysis. Although product ions with same molecular formulas might lead to low correlation (See details in section 3.3) and would suggest different species (i.e., isomers, fragment ions,...), a few molecules with $R^2$ greater than 0.9 (18 for $NO_3^-$-LTOF, 32 for PTR-TOF, and 5 for $I^-$-CIMS) were selected and be likely attributed to the same species.*

*Although no direct calibration has been performed for the $NH_4^+$-Orbitrap, the OOM concentrations were estimated based on comparisons between $NH_4^+$-Orbitrap and the two reference instruments which has developed reliable quantification methods. For the OOMs whose timeseries had $R^2$ greater than 0.9 between $NH_4^+$-Orbitrap and REF, linear regression was conducted for normalized intensities in $NH_4^+$-Orbitrap (dimensionless) and concentrations in REF (molecules cm$^{-3}$), and the slopes were recorded as their relative sensitivity. The calibration factor $c_{Orbi-REF}$ was derived from the averaged relative sensitivity of these species (~2.62 × 10$^8$ molecules cm$^{-3}$ for $NO_3^-$-LTOF and ~4.83 × 10$^8$ molecules cm$^{-3}$ for PTR3-TOF). Applying the calibration factors to all the OOMs, their concentrations detected by $NH_4^+$-Orbitrap could be calculated as shown in equation (4). Additionally, a temperature-dependent sampling-line loss correction factor was applied (Simon et al., 2020).*

$$[OOM]_{Orbi-REF} = c_{Orbi-REF} \times [OOM]_{nor} \qquad (4)$$

Lines 343-357: *The reason why the correlations of certain molecules are lower than 0.9 might be due to the molecules' composition or potential ionization artifacts. RH dependence is an important property leading to the low correlations as the experiment includes RH variation from 20% to 80%. Although $NO_3^-$ ion chemistry had been reported to be less dependent on RH (Viggiano et al., 1997), the sensitivities of PTR3-TOF (Breitenlechner et al., 2017) and $I^-$-CIMS (Lee et al., 2014) both showed high dependence on RH. In addition, the relative sensitivity of $NH_4^+$-Orbitrap was also influenced by the varying RH (See details in Section 3.6). Fragmentation, such as decomposition of dimers, would also lead to low correlations. However, less fragmentation is expected to occur in the $NH_4^+$-Orbitrap using similar settings as our earlier studies (Riva et al., 2019a; Riva et al., 2020). In comparison, decomposition of peroxides (i.e., ROOR and ROOH) can be expected within the PTR3-TOF. While fragmentation of dimeric compounds can contribute to the overall signal of the monomers, the concentration of such species remains minor (Li et al., 2022). As a result, no large enhancement of the monomers signal intensity is expected. There are also other artifacts which cannot be excluded based on current dataset, including potential isomers and differences in response time between instruments, would also lead to the low correlations.*

NO3-CIMS and PTR-MS are both more selective than NH4+-CIMS.

In this study, the PTR3-TOF had been optimized for amine detection, while in a previous study, PTR3-TOF has been shown to measure molecules with oxygen number ranging from 0 to 18 (Breitenlechner et al., 2017). As result, we do not want to conclude on the selective of the different ionization techniques used in this work. Overall, we believe that the $NH_4^+$-Orbitrap has a lower selectivity compared to current $NO_3^-$-CIMS (sensitive to highly oxidized molecules) and traditional PTR-MS (sensitive to molecules with lower oxygen numbers), according to our previous study (Riva et al., 2019b).

2. In several occasions, the authors responded to my questions (e.g., #10 and a couple of others), but the clarification was not added to the revised manuscript.

We added revisions for comments #10 in this version. But for other questions (i.e., #5 and #13), we think there is no proper position to add the clarification in the paper, thus we only replied to the reviewer without revising the manuscript or SI. We would like to point out that the discussion will be part of the article and will remain available.

For #10, the meaning of values following "±" are clarified in the revised manuscript.

For #5, as we have no concrete evidence from Vocus-$NH_4^+$ or other $NH_4^+$-CIMS to make a comparison, we did not mention it within the manuscript for the discussion.

For #13, we have added some discussions regarding the uncertainty i.e., Lines 342-355. As we have explained, there are multiple factors which might affect the correlation. Showing the elemental characteristics of one or several species to the readers would not summarize the real situations and might lead to unnecessary misunderstanding.

Lines 311-314: *The number of O atoms in OOMs varied from 1 to 11 in monomers ($C_{2-10}$) and from 2 to 16 for dimeric products ($C_{14-20}$), with an average elemental oxygen-to-carbon ratio (O:C) of 0.4 ± 0.2 (the value following "±" herein and after refers to the standard deviation of O:C during the experiment).*

3. Without the calibration or estimation of sensitivity in any way, I suggest not use the terminology "semi-quantitative". At best, this is intensity-based relative comparison among different instruments.

We have revised the terminology in the different parts of the manuscript.

Lines 42-44: *OOMs concentrations measured by $NH_4^+$-Orbitrap were estimated using calibration factors derived from the OOMs with high timeseries correlations during the side-by-side measurements.*

Lines 361-362: *The sensitivity of $NH_4^+$-Orbitrap was constrained based on the intensity comparison between $NH_4^+$-Orbitrap and the other two instruments.*

Lines 369-371: *Taking into consideration that such ranges are also the oxygen number ranges with high sensitivities respectively, this proves the robustness of the $NH_4^+$-Orbitrap and the intensity-based relative comparison between $NH_4^+$-Orbitrap and two reference instruments.*

Breitenlechner, M., Fischer, L., Hainer, M., Heinritzi, M., Curtius, J., and Hansel, A.: PTR3: An Instrument for Studying the Lifecycle of Reactive Organic Carbon in the Atmosphere, Analytical Chemistry, 89, 5824-5831, 10.1021/acs.analchem.6b05110, 2017.

Lee, B. H., Lopez-Hilfiker, F. D., Mohr, C., Kurtén, T., Worsnop, D. R., and Thornton, J. A.: An Iodide-Adduct High-Resolution Time-of-Flight Chemical-Ionization Mass Spectrometer: Application to Atmospheric Inorganic and Organic Compounds, Environmental Science & Technology, 48, 6309-6317, 10.1021/es500362a, 2014.

Riva, M., Brüggemann, M., Li, D., Perrier, S., George, C., Herrmann, H., and Berndt, T.: Capability of CI-Orbitrap for Gas-Phase Analysis in Atmospheric Chemistry: A Comparison with

the CI-APi-TOF Technique, Analytical Chemistry, 92, 8142-8150, 10.1021/acs.analchem.0c00111, 2020.

Riva, M., Ehn, M., Li, D., Tomaz, S., Bourgain, F., Perrier, S., and George, C.: CI-Orbitrap: An Analytical Instrument To Study Atmospheric Reactive Organic Species, Analytical chemistry, 2019a.

Riva, M., Rantala, P., Krechmer, J. E., Peräkylä, O., Zhang, Y., Heikkinen, L., Garmash, O., Yan, C., Kulmala, M., Worsnop, D., and Ehn, M.: Evaluating the performance of five different chemical ionization techniques for detecting gaseous oxygenated organic species, Atmospheric Measurement Techniques, 12, 2403-2421, 10.5194/amt-12-2403-2019, 2019b.

Viggiano, A. A., Seeley, J. V., Mundis, P. L., Williamson, J. S., and Morris, R. A.: Rate Constants for the Reactions of $XO_3^-(H_2O)_n$ (X = C, HC, and N) and $NO_3^-(HNO_3)n$ with $H_2SO_4$: Implications for Atmospheric Detection of $H_2SO_4$, The Journal of Physical Chemistry A, 101, 8275-8278, 10.1021/jp971768h, 1997.